# Learning Disentangled Representations with Reference-Based Variational Autoencoders

## Abstract

Learning disentangled representations from visual data, where different high-level generative factors are independently encoded, is of importance for many computer vision tasks. Supervised approaches, however, require a significant annotation effort in order to label the factors of interest in a training set. To alleviate the annotation cost, we introduce a learning setting which we refer to as "reference-based disentangling". Given a pool of unlabelled images, the goal is to learn a representation where a set of target factors are disentangled from others. The only supervision comes from an auxiliary "reference set" that contains images where the factors of interest are constant. In order to address this problem, we propose reference-based variational autoencoders, a novel deep generative model designed to exploit the weak supervisory signal provided by the reference set. During training, we use the variational inference framework where adversarial learning is used to minimize the objective function. By addressing tasks such as feature learning, conditional image generation or attribute transfer, we validate the ability of the proposed model to learn disentangled representations from minimal supervision.

## 1 Introduction

Natural images can be considered the result of a generative process involving many factors of variation. For instance, the appearance of a face is determined by the interaction between many latent variables including the pose, the illumination, the subject's age and expression. Given that the interaction between these underlying explanatory factors is usually very complex, inverting the generative process is extremely challenging. From this perspective, learning disentangled representations where different high-level generative factors are independently encoded, can be considered one of the most relevant problems in computer vision (Bengio et al., 2013). For instance, these representations can be applied to complex classification tasks given that features correlated with the image labels can be identified. We can find another example in conditional image generation (van den Oord et al., 2016; Yan et al., 2016), where disentangled representations allow to manipulate desired attributes in synthesized images.

**Motivation:** Latent variable models have a long history as tools to learn abstract data representations. The main idea behind these methods is to define a probabilistic model relating the observations, *e.g.* images, with a set of latent variables encoding the factors of variation underlying the data. In recent years, Variational autoencoders (VAEs) (Kingma & Welling, 2014) have emerged as a powerful latent variable model coupling deep learning with variational inference. However, even though VAEs are able to build generic data representations, they are trained in an unsupervised manner and, thus, they lack a mechanism to impose specific high-level semantics on the latent variables. In order to address this limitation, different semi-supervised autoencoders have been proposed (Rifai et al., 2012; Kingma et al., 2014). In these approaches, a subset of the latent factors are assumed to be manually labelled in a training set. These annotations provide supervision to the model, and allow to disentangle the labelled variables from the remaining generative factors. The main drawback of this strategy is that it require a significant effort in order to annotate the targeted factors in a dataset.

In this context, we are motivated by the following question: "Is it possible to disentangle specific factors of variation with minimal supervision?".

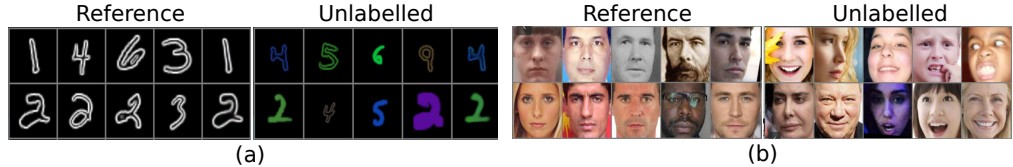

Figure 1: Illustration of different reference-based disentangling problems. (a) Disentangling style from digits. The reference distribution is composed by numbers with a fixed style (b) Disentangling factors of variations related with facial expressions. Reference images correspond to neutral faces.

**Contributions.** In this paper, we introduce "reference-based disentangling". A learning setting in which, given a training set of unlabelled images, the goal is to learn a representation where a specific set of generative factors are disentangled from the rest. For that purpose, supervision comes in the form of an auxiliary "reference set" containing images where the factors of interest are constant. See Fig. 1 for illustrative examples. Different from a semi-supervised scenario, explicit labels are not available during training. In contrast, reference-based disentangling is a weakly-supervised task, where the reference set only provides implicit information about the generative factors that we aim to disentangle. Note that a collection of reference images is generally easier to obtain compared to explicitly annotations of target factors. For instance, if we are interested in disentangling facial gesture information from faces, we would need to annotate images according to different expression classes. Despite the fact that this is feasible for a reduced number of basic gestures, naturalistic expressions depend on a combination of a large number of facial muscle activations with their corresponding intensities (Ekman & Rosenberg, 1997). As a consequence, annotating facial expression datasets typically requires a very expensive process. By contrast, collecting a large collection of neutral faces is much easier and can be carried out by non-expert annotators.

The main contributions of our paper are summarized as follows:

- We propose reference-based variational autoencoders (Rb-VAEs). Different from standard unsupervised VAEs, our model is able to impose high-level semantics into the latent variables by exploiting the weak supervision provided by the reference set.

- We identify critical limitations of the VAE objective function when used to train our model. For this reason, we propose an alternative training procedure based on recently introduced ideas in the context of variational inference and adversarial learning.

- By learning disentangled representations from minimal supervision, we show how the proposed framework is able to naturally address different tasks such as feature learning, conditional image generation, and attribute transfer.

## 2    RELATED WORK

**Deep Generative Models** have been extensively explored to model visual and other types of data during recent years. Variational autoencoders (Kingma & Welling, 2014) and generative adversarial networks (GANs) (Goodfellow et al., 2014) have emerged as two of the most effective frameworks. VAEs use variational inference in order to learn an encoder network that maps images to a posterior distribution over latent variables. Similarly, a decoder network is learned that produces the conditional distribution on images given the latent variables. GANs are also composed of two differentiable networks. The generator network synthesizes images from latent variables, similar to the VAE decoder. The discriminator network discriminates between real training images and generated synthetic images. During training, GANs employ an adversarial learning procedure which allows to simultaneously optimize the discriminator and generator parameters. Even though GANs have been shown to generate more realistic samples than VAEs, their main limitation is the lack of an inference mechanism able to map images into their corresponding latent variables. In order to address this drawback, there have been several attempts to combine ideas from VAEs and GANs (Larsen et al., 2015; Dumoulin et al., 2017; Donahue et al., 2017). Interestingly, it has been shown that adversarial learning can be used to minimize the variational objective function of VAEs (Makhzani et al., 2016; Huszár, 2017). Inspired by this observation, various methods such as adversarial vari-

ational Bayes (Mescheder et al., 2017), $\alpha$-GAN (Rosca et al., 2017), and symmetric-VAE (sVAE) (Pu et al., 2018) have incorporated adversarial learning into the VAE framework.

Different from this prior work, our Rb-VAE model is a deep generative model specifically designed to solve the reference-based disentangling problem. During training, adversarial learning is used in order to minimize a variational objective function inspired by the one employed in sVAE (Pu et al., 2018). Although sVAE was originally motivated by the limitations of the maximum likelihood criterion used in unsupervised VAE, we show how the variational formulation of sVAE offers specific advantages in the context of our proposed model.

**Learning disentangled representations** is a long standing problem in machine learning and computer vision (Bengio et al., 2013). In the literature, we can differentiate three main paradigms to address it: unsupervised, supervised, and weakly-supervised. Unsupervised models are trained without specific information about the generative factors of interest (Desjardins et al., 2012; Chen et al., 2016). To address this task, the most common approach consists in imposing different constraints on the latent representation. For instance, unsupervised VAEs typically define the prior over the latent variables with a fully-factorized Gaussian distribution. Given that high-level generative factors are typically independent, this prior encourage their disentanglement in different dimensions of the latent representation. Based on this observation, different approaches such as $\beta$-VAE (Higgins et al., 2017), DIP-VAE (Kumar et al., 2018), FactorVAE Kim & Mnih (2018) or $\beta$-TCVAE (Chen et al., 2018) have explored more sophisticated regularization mechanisms over the distribution of inferred latent variables. Although unsupervised approaches are able to identify simple explanatory variables of visual data, these methods do not allow latent variables to model specific high-level factors of variation.

A straight-forward approach to overcome this limitation is to use a fully-supervised strategy. In this scenario, models are learned by using a training set where target factors of interest are explicitly labelled. Following this paradigm, we can find different semi-supervised (Kingma et al., 2014; Narayanaswamy et al., 2017), and conditional (Yan et al., 2016; Pu et al., 2016) variants of autoencoders. In spite of the effectiveness of supervised approaches in different applications (Ma et al., 2017; Pu et al., 2016; Tran et al., 2017), obtaining explicit labels is not feasible in scenarios where we aim to disentangle a large number of factors or their annotation is difficult.

An intermediate solution between unsupervised and fully-supervised methods are weakly-supervised approaches. In this case, only implicit information about factors of variation is provided during training. Several works have explored this strategy by using different forms of weak-supervision such as: temporal coherence in sequential data (Hsu et al., 2017; Denton et al., 2017; Villegas et al., 2017), pairing relations between images sharing the same generative factors (Mathieu et al., 2016; Donahue et al., 2018), grouping information for training samples (Bouchacourt et al., 2017), or partial knowledge about the rendering process in computer graphics (Yang et al., 2015; Kulkarni et al., 2015).

Other than previous approaches relying on other forms of weak supervision, our proposed method addresses the reference-based disentangling problem. In this scenario, the challenge is to exploit the implicit information provided by a training set of images where the generative factors of interest are constant, *e.g.* a set of faces with neutral expression. Note that for this problem, supervised approaches are not a directly applicable since explicit labels are not available.

## 3 PRELIMINARIES: VARIATIONAL AUTOENCODERS

Variational autoencoders (VAEs) (Kingma & Welling, 2014) are generative models defining a joint distribution $p_\theta(\mathbf{x}, \mathbf{z}) = p_\theta(\mathbf{x}|\mathbf{z})p(\mathbf{z})$, where $\mathbf{x}$ is an observation, *e.g.* an image, and $\mathbf{z}$ is a latent variable with a simple prior $p(\mathbf{z})$, *e.g.* a Gaussian with zero mean and identity covariance matrix. Moreover, $p_\theta(\mathbf{x}|\mathbf{z})$ is typically modeled as a factored Gaussian, whose mean and diagonal covariance matrix are given by a non-linear function of $\mathbf{z}$, implemented as a deep (convolutional) neural network. The latter network is referred to as the "decoder", or "generator".

Given a training set of i.i.d. samples from an unknown data distribution $p(\mathbf{x})$, the goal of VAEs is two-fold: *(i)* To learn the optimal generator parameters $\theta$ so that is possible to synthesize samples following $p(\mathbf{x})$ from the prior $p(\mathbf{z})$, *(ii)*: To approximate the intractable posterior $p_\theta(\mathbf{z}|\mathbf{x})$ in order to infer latent variables $\mathbf{z}$ from a given observation $\mathbf{x}$. For this purpose, VAEs define a variational

distribution $q_\psi(\mathbf{x}, \mathbf{z}) = q_\psi(\mathbf{z}|\mathbf{x})p(\mathbf{x})$. The approximate posterior $q_\psi(\mathbf{z}|\mathbf{x})$ is defined as another factored Gaussian, whose mean and diagonal covariance matrix are given as the output of an "encoder" or "inference" network with parameters $\psi$ and taking $\mathbf{x}$ as an input. During training, the parameters of the decoder and encoder are learned jointly by solving the following optimization problem:

$$\min_{\theta, \psi} \mathbb{E}_{p(\mathbf{x})} \Big[ \mathbb{KL}(q_\psi(\mathbf{z}|\mathbf{x}) \parallel p(\mathbf{z})) - \mathbb{E}_{q_\psi(\mathbf{z}|\mathbf{x})} \log(p_\theta(\mathbf{x}|\mathbf{z})) \Big], \quad (1)$$

which is equivalent to the minimization of the $\mathbb{KL}$ divergence between $q_\psi(\mathbf{x}, \mathbf{z})$ and $p_\theta(\mathbf{x}, \mathbf{z})$. Note that the first term can be interpreted as a regularization mechanism encouraging the distribution $q_\psi(\mathbf{z}|\mathbf{x})$ to be similar to the prior $p(\mathbf{z})$. The second term is known as the reconstruction error, measuring the negative log-likelihood of a generated sample $\mathbf{x}$ from its inferred latent variables $q_\psi(\mathbf{z}|\mathbf{x})$. The described minimization problem is usually solved by using stochastic gradient descent (SGD) where $p(\mathbf{x})$ is approximated by the training set. The "re-parametrization trick" (Rezende et al., 2014) is used to enable gradient back-propagation across samples from $q_\phi(\mathbf{z}|\mathbf{x})$.

# 4 REFERENCE-BASED DISENTANGLED REPRESENTATIONS

Consider a training set of unlabelled images (*e.g.* human faces) $\mathbf{x} \in \mathbb{R}^{W \times H \times 3}$ sampled from a given distribution $p^u(\mathbf{x})$. Our goal is to learn a latent variable model defining a joint distribution over $\mathbf{x}$ and latent variables $\mathbf{e} \in \mathbb{R}^{D_e}$ and $\mathbf{z} \in \mathbb{R}^{D_z}$. Whereas $\mathbf{e}$ is expected to encode information about a set of generative factors of interest, *e.g.* facial expressions, $\mathbf{z}$ should model the remaining factors of variation underlying the images, *e.g.* pose, illumination, age, etc. From now on, we will refer to $\mathbf{e}$ and $\mathbf{z}$ as the "target" and "common factors", respectively. In order to disentangle them, we are provided with an additional set of reference images sampled from $p^r(\mathbf{x})$, representing a distribution over $\mathbf{x}$ where target factors $\mathbf{e}$ are constant *e.g.* neutral faces. Given $p^r(\mathbf{x})$ and $p^u(\mathbf{x})$, we define an auxiliary binary variable $y \in \{0, 1\}$ indicating whether an image $\mathbf{x}$ has been sampled from the unlabelled or reference distributions, *i.e.* $p(\mathbf{x}|y = 0) = p^u(\mathbf{x})$ and $p(\mathbf{x}|y = 1) = p^r(\mathbf{x})$. In reference-based disentangling we aim to exploit the weak-supervision provided by $y$ in order to effectively disentangle target and common factors.

## 4.1 REFERENCE-BASED VARIATIONAL AUTOENCODERS

In this section, we present our reference-based variational autoencoder (Rb-VAE): a deep latent variable model that defines a joint distribution $p_\theta(\mathbf{x}, \mathbf{z}, \mathbf{e}, y)$ as

$$p_\theta(\mathbf{x}, \mathbf{z}, \mathbf{e}, y) = p_\theta(\mathbf{x}|\mathbf{z}, \mathbf{e})p(\mathbf{z})p(\mathbf{e}|y)p(y), \quad (2)$$

where conditional dependencies are designed to explicitly address the reference-based disentangling problem. See Figure 2 for a schematic illustration of our model. We define $p_\theta(\mathbf{x}|\mathbf{z}, \mathbf{e}) = \mathcal{L}(\mathbf{x}|\mathcal{G}_\theta(\mathbf{z}, \mathbf{e}), \lambda)$, where $\mathcal{G}_\theta(\mathbf{z}, \mathbf{e})$ is a decoder network that maps a pair of latent variables $(\mathbf{z}, \mathbf{e})$ to an image defining the mean of a Laplace distribution $\mathcal{L}$ with fixed scale parameter $\lambda$. Note that we use a Laplace distribution instead of the Gaussian usually employed in the VAEs. The reason is that the negative log-likelihood is equivalent to the $\ell_1$-loss which is known to encourage sharper image reconstructions with better visual quality.

To reflect the assumption of constant target factors across reference images, we define the conditional distribution over $\mathbf{e}$ given $y = 1$ as a delta peak centered on a learned vector $\mathbf{e}^r \in R^{D_e}$, *i.e.* $p(\mathbf{e}|y = 1) = \delta(\mathbf{e} - \mathbf{e}^r)$. In contrast, for $y = 0$, the conditional distribution is set to a unit Gaussian: $p(\mathbf{e}|y = 0) = \mathcal{N}(\mathbf{e}|\mathbf{0}, \mathbf{I})$. In the following, we denote $p(\mathbf{e}|y = 0) = p(\mathbf{e})$. Contrary to the case of target factors $\mathbf{e}$, the prior over common factors $\mathbf{z}$ is equal for reference and unlabelled images, and taken to be a unit Gaussian $y$ for $p(\mathbf{z}) = \mathcal{N}(\mathbf{z}|\mathbf{0}, \mathbf{I})$. Finally, we assume a uniform prior over $y$, *i.e.* $p(y = 0) = p(y = 1) = \frac{1}{2}$.

## 4.2 CONVENTIONAL VARIATIONAL LEARNING

Following the VAE framework described in Section 3, we can define a variational distribution $q_\psi(\mathbf{z}, \mathbf{e}, \mathbf{x}, y) = q_\psi(\mathbf{z}|\mathbf{x})q_\psi(\mathbf{e}|\mathbf{x}, y)p(\mathbf{x}, y)$, and learn the model parameters $\theta$ by minimizing the $\mathbb{KL}$ divergence between $q_\psi$ and $p_\theta$. In this case, note that the conditionals $q_\psi(\mathbf{e}|\mathbf{x}, y)$ and $q_\psi(\mathbf{z}|\mathbf{x})$ provide a factored approximation of the intractable posterior $p_\theta(\mathbf{e}, \mathbf{z}|\mathbf{x}, y)$, allowing to infer target and common factors $\mathbf{e}$ and $\mathbf{z}$ given an image $\mathbf{x}$. Given a reference image, *i.e.* with $y = 1$,

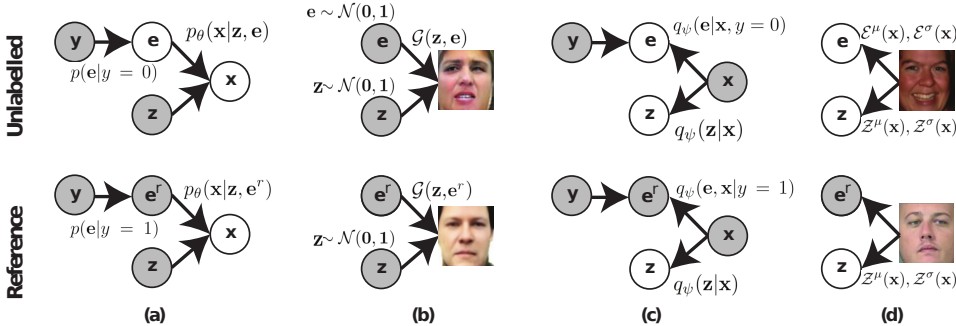

Figure 2: Overview of the proposed Rb-VAE model. Shaded circles correspond to observed variables in each scenario. (a) Generative process where $p_\theta(\mathbf{x}|\mathbf{z}, \mathbf{e})$ maps latent variables $\mathbf{z}$ (common factors) and $\mathbf{e}$ (target factors) to images $\mathbf{x}$. Reference images are known to be generated by constant $\mathbf{e}^r$. (b) The decoder is a deep network $\mathcal{G}$ with inputs $\mathbf{z}, \mathbf{e}$. (c) Encoders map images $\mathbf{x}$ to the corresponding common and target factors $\mathbf{z}$ and $\mathbf{e}$ respectively. (d) The encoders $\mathcal{E}$ and $\mathcal{Z}$ are implemented as deep nets modelling two Gaussians distribution over $\mathbf{e}$ and $\mathbf{z}$ respectively.

the target factors $q_\psi(\mathbf{e}|\mathbf{x}, y = 1)$ are known to be equal to the reference value $\mathbf{e}^r$. On the other hand, given an non-reference image, *i.e.* with $y = 0$, we define the approximate posterior $q_\psi(\mathbf{e}|\mathbf{x}, y = 0) = \mathcal{N}(\mathbf{e}|\mathcal{E}^\mu(\mathbf{x}), \mathcal{E}^\sigma(\mathbf{x}))$, where the means and diagonal covariance matrices of a conditional Gaussian distribution are given by non-linear functions $\mathcal{E}^\mu(\mathbf{x})$ and $\mathcal{E}^\sigma(\mathbf{x})$. Similarly, we use an additional network to model $q_\psi(\mathbf{z}|\mathbf{x}) = \mathcal{N}(\mathbf{z}|\mathcal{Z}^\mu(\mathbf{x}), \mathcal{Z}^\sigma(\mathbf{x}))$.

**Optimization with SGVB:** In Appendix (A) we show that the minimization of the $\mathbb{KL}$ divergence between $q_\psi$ and $p_\theta$ can be expressed as:

$$\min_{\theta, \psi, \mathbf{e}^r} \quad \mathbb{E}_{p^u(\mathbf{x})} \Big[ \mathbb{KL}(q_\psi(\mathbf{z}|\mathbf{x})q_\psi(\mathbf{e}|\mathbf{x}) \parallel p(\mathbf{z})p(\mathbf{e})) - \mathbb{E}_{q_\psi(\mathbf{z}|\mathbf{x})q_\psi(\mathbf{e}|\mathbf{x})} \log(p_\theta(\mathbf{x}|\mathbf{z}, \mathbf{e})) \Big]$$
$$+ \mathbb{E}_{p^r(\mathbf{x})} \Big[ \mathbb{KL}(q_\psi(\mathbf{z}|\mathbf{x}) \parallel p(\mathbf{z})) - \mathbb{E}_{q_\psi(\mathbf{z}|\mathbf{x})} \log(p_\theta(\mathbf{x}|\mathbf{z}, \mathbf{e}^r)) \Big], \tag{3}$$

where the second and fourth terms of the expression correspond to the reconstruction errors for unlabelled and reference images respectively. Note that for reference images, no inference over target factors $\mathbf{e}$ is needed. Instead, the generator reconstructs them using the optimized parameter $\mathbf{e}^r$. Similar to standard VAEs, the remaining terms consist of $\mathbb{KL}$ divergences between approximate posteriors and priors over the latent variables. The minimization problem defined in Eq. (3) can be solved using SGD and the *re-parametrization* trick in order to back-propagate the gradient when sampling from $q_\psi(\mathbf{e}|\mathbf{x})$ and $q_\psi(\mathbf{z}|\mathbf{x})$.

### 4.3 VARIATIONAL LEARNING WITH SYMMETRIC KL DIVERGENCE

The main limitation of the variational objective defined in Eq. (3) is that it does not guarantee that common and target factors will be effectively disentangled in $\mathbf{z}$ and $\mathbf{e}$ respectively. In order to understand this phenomenon, it is necessary to analyze the role of the conditional prior $p(\mathbf{e}|y)$ in Rb-VAEs. By defining $p(\mathbf{e}|y = 1)$ as a delta function, the model is forced to encode into $\mathbf{z}$ all the generative factors of reference images, given that they must be reconstructed via $p_\theta(\mathbf{x}|\mathbf{z}, \mathbf{e}^r)$ with constant $\mathbf{e}^r$. Therefore, $p(\mathbf{e}|y)$ is implicitly encouraging $q_\psi(\mathbf{z}|\mathbf{x})$ to encode common factors present in reference and unlabelled samples. However, this mechanism does not avoid the scenario where target factors are also encoded into latent variables $\mathbf{z}$. More formally, the minimization of (3) does not prevent a degenerate solution $p_\theta(\mathbf{x}|\mathbf{z}, \mathbf{e}) = p_\theta(\mathbf{x}|\mathbf{z})$, where the inferred latent variables by $q_\psi(\mathbf{e}|\mathbf{x})$ are uninformative and the decoder ignores them.

In order to address this limitation, we propose to optimize an alternative variational expression inspired by unsupervised Symmetric VAEs (Pu et al., 2018). Specifically, we add into the minimized objective the reversed $\mathbb{KL}$ between $q_\psi$ and $p_\theta$ as:

$$\min_{\theta, \psi} \quad \mathbb{KL}(q_\psi(\mathbf{z}, \mathbf{e}, \mathbf{x}, y) \parallel p_\theta(\mathbf{x}, \mathbf{z}, \mathbf{e}, y)) + \mathbb{KL}(p_\theta(\mathbf{x}, \mathbf{z}, \mathbf{e}, y) \parallel q_\psi(\mathbf{z}, \mathbf{e}, \mathbf{x}, y)), \tag{4}$$

In order to understand why this additional term allows to mitigate the degenerate solution $p_\theta(\mathbf{x}|\mathbf{z}, \mathbf{e}) = p_\theta(\mathbf{x}|\mathbf{z})$, it is necessary to observe that its minimization is equivalent to:

$$\min_{\theta, \psi} \quad \mathbb{E}_{p(\mathbf{z}, \mathbf{e})} \Big[ \mathbb{KL}(p_\theta(\mathbf{x}|\mathbf{z}, \mathbf{e}) \parallel p^u(\mathbf{x})) - \mathbb{E}_{p_\theta(\mathbf{x}|\mathbf{z}, \mathbf{e})} [\log(q_\psi(\mathbf{z}|\mathbf{x})) + \log(q_\psi(\mathbf{e}|\mathbf{x}))] \Big]$$

$$+ \mathbb{E}_{p(\mathbf{z})p_\theta(\mathbf{x}|\mathbf{z}, \mathbf{e}^r)} \Big[ \mathbb{KL}(p_\theta(\mathbf{x}|\mathbf{z}, \mathbf{e}^r) \parallel p^r(\mathbf{x})) - \log(q_\psi(\mathbf{z}|\mathbf{x})) \Big], \quad (5)$$

see Appendix A for a detailed derivation. In the defined expression, the two $\mathbb{KL}$ divergences encourage images generated using $p(\mathbf{z})$, $p(\mathbf{e})$ and $\mathbf{e}^r$ to be similar to samples from the real distributions $p^r(\mathbf{x})$ and $p^u(\mathbf{x})$. On the other hand, the remaining terms correspond to reconstruction errors over latent variables $\mathbf{z}, \mathbf{e}$ inferred from generated images drawn from $p_\theta$. As a consequence, the minimization of these errors is encouraging the decoder $p_\theta(\mathbf{x}|\mathbf{z}, \mathbf{e})$ to generate images $\mathbf{x}$ by taking into account latent variables $\mathbf{e}$, since the latter must be reconstructed via $q_\psi(\mathbf{e}|\mathbf{x})$. In conclusion, the addition of the reversed $\mathbb{KL}$ in the objective avoids the degenerate solution that ignores variables $\mathbf{e}$.

**Optimization via Adversarial Learning.** Given the introduction of the reversed KL divergence, the learning procedure described in Section 4.2 can not be directly applied to the minimization of Eq. (4). However, note that the objective function defined in Eq. (4) is equivalent to:

$$\mathbb{E}_{q_\psi(\mathbf{e}, \mathbf{z}|\mathbf{x})p^u(\mathbf{x})} \log \left( \frac{q_\psi(\mathbf{e}, \mathbf{z}|\mathbf{x})p^u(\mathbf{x})}{p_\theta(\mathbf{x}|\mathbf{e}, \mathbf{z})p(\mathbf{z}, \mathbf{e})} \right) + \mathbb{E}_{p_\theta(\mathbf{x}|\mathbf{e}, \mathbf{z})p(\mathbf{z}, \mathbf{e})} \log \left( \frac{p_\theta(\mathbf{x}|\mathbf{e}, \mathbf{z})p(\mathbf{z}, \mathbf{e})}{q_\psi(\mathbf{e}, \mathbf{z}|\mathbf{x})p^u(\mathbf{x})} \right) +$$

$$\mathbb{E}_{q_\psi(\mathbf{z}|\mathbf{x})p^r(\mathbf{x})} \log \left( \frac{q_\psi(\mathbf{z}|\mathbf{x})p(\mathbf{x})^r}{p_\theta(\mathbf{x}|\mathbf{e}^r, \mathbf{z})p(\mathbf{z})} \right) + \mathbb{E}_{p_\theta(\mathbf{x}|\mathbf{e}^r, \mathbf{z})p(\mathbf{z})} \log \left( \frac{p_\theta(\mathbf{x}|\mathbf{e}^r, \mathbf{z})p(\mathbf{z})}{q_\psi(\mathbf{z}|\mathbf{x})p^r(\mathbf{x})} \right), \quad (6)$$

corresponding to the minimization of: *(i)* A difference of two expectations over the log-density ratio of $q_\psi(\mathbf{e}, \mathbf{z}|\mathbf{x})p^u(\mathbf{x})$ and $p_\theta(\mathbf{x}|\mathbf{e}, \mathbf{z})p(\mathbf{z})p(\mathbf{e})$. *(ii)* An analogous expression for distributions $q_\psi(\mathbf{z}|\mathbf{x})p^r(\mathbf{x})$ and $p_\theta(\mathbf{x}|\mathbf{e}^r, \mathbf{z})p(\mathbf{z}))$. By approximating both density ratios with two auxiliary functions:

$$d_\xi(\mathbf{x}, \mathbf{z}, \mathbf{e}) = \log \left( \frac{q_\psi(\mathbf{e}, \mathbf{z}|\mathbf{x})p^u(\mathbf{x})}{p_\theta(\mathbf{x}|\mathbf{e}, \mathbf{z})p(\mathbf{z})p(\mathbf{e})} \right), \ d_\gamma(\mathbf{x}, \mathbf{z}) = \log \left( \frac{q_\psi(\mathbf{z}|\mathbf{x})p^r(\mathbf{x})}{p_\theta(\mathbf{x}|\mathbf{e}^r, \mathbf{z})p(\mathbf{z})} \right). \quad (7)$$

we can express the minimization problem in Eq. (6) as:

$$\min_{\theta, \psi} \quad \mathbb{E}_{q_\psi(\mathbf{e}, \mathbf{z}|\mathbf{x})p^u(\mathbf{x})} d_\xi(\mathbf{x}, \mathbf{z}, \mathbf{e}) - \mathbb{E}_{p_\theta(\mathbf{x}|\mathbf{e}, \mathbf{z})p(\mathbf{z})p(\mathbf{e})} d_\xi(\mathbf{x}, \mathbf{z}, \mathbf{e})$$

$$+ \mathbb{E}_{q_\psi(\mathbf{z}|\mathbf{x})p^r(\mathbf{x})} d_\gamma(\mathbf{x}, \mathbf{z}) - \mathbb{E}_{p_\theta(\mathbf{x}|\mathbf{e}^r, \mathbf{z})p(\mathbf{z})} d_\gamma(\mathbf{x}, \mathbf{z}) \quad (8)$$

which can be solved with SGD. Concretely, we can evaluate functions $d_\xi(\mathbf{x}, \mathbf{z}, \mathbf{e})$, $d_\gamma(\mathbf{x}, \mathbf{z})$ and back-propagate the gradients w.r.t parameters $\psi$ and $\theta$ by using the *re-parametrization trick* over samples of $\mathbf{x}, \mathbf{e}$ and $\mathbf{z}$.

The main challenge of approximating the log-density ratio using the functions defined in Eq. (7), is that parameters $\xi$ and $\gamma$ are not known and must be also optimized. Fortunately, the log-ratio between two distributions can be estimated by using logistic regression (Bickel et al., 2009). For instance, given fixed encoder and generator parameters $\theta$ and $\psi$, we can optimize $\xi$ by solving:

$$\max_{\xi} \ \mathbb{E}_{p_\theta(\mathbf{x}|\mathbf{z}, \mathbf{e})p(\mathbf{z}, \mathbf{e})} \log(\sigma(d_\xi(\mathbf{x}, \mathbf{z}, \mathbf{e})) + \mathbb{E}_{q_\psi(\mathbf{e}, \mathbf{z}|\mathbf{x})p^u(\mathbf{x})} \log(1 - \sigma(d_\xi(\mathbf{x}, \mathbf{z}, \mathbf{e})), \quad (9)$$

where $\sigma(\cdot)$ refers to the sigmoid function. A similar strategy can be applied to optimize parameters of $d_\gamma(\mathbf{x}, \mathbf{z})$. This approach is analogous to adversarial unsupervised methods such as ALI (Dumoulin et al., 2017), where the function $d_\gamma(\cdot)$ acts as a discriminator trying to distinguish whether pairs of reference images $\mathbf{x}$ and latent variables $\mathbf{z}$ generated by $q_\psi$ and $p_\theta$. However, in our case we have an additional discriminator $d_\xi$ operating over unlabelled images and its corresponding latent variables $\mathbf{z}$ and $\mathbf{e}$. Interestingly, note that $d_\xi$ encourages the distribution $q_\psi(\mathbf{z}, \mathbf{e}|x)$ to be similar to the prior $p(\mathbf{z}, \mathbf{e})$. Therefore, it forces representations $\mathbf{z}$ and $\mathbf{e}$ to be conditionally independent. This is especially interesting in our context since we assume that target and common factors are uncorrelated. To conclude, it is also interesting to observe that the discriminator $d_\gamma(\mathbf{x}, \mathbf{z})$ is implicitly encouraging latent variables $\mathbf{z}$ to encode only information about the common factors. The reason is that samples generated from $p_\theta(\mathbf{x}|\mathbf{z}, \mathbf{e^r})p(\mathbf{z})$ are forced to be similar to reference images. As a consequence, z can not contain information about target factors, which must be encoded into e.

Using previous definitions, we use an adversarial procedure where model and discriminators parameters $(\theta, \psi)$, and $(\xi, \gamma)$ are simultaneously optimized by minimizing and maximizing Eqs. (8) and (9) respectively. The algorithm used to process one batch during SGD is shown in Appendix B. In Rb-VAEs, the discriminators $d_\gamma(\cdot)$ and $d_\xi(\cdot)$ are also implemented as deep convolutional networks.

**Explicit log-likelihood maximization.** As shown in Eqs. (3) and (5), the minimization of the symmetric $\mathbb{KL}$ divergence encourages low reconstruction errors for images and inferred latent variables. However, by using the proposed adversarial learning procedure, the minimization of these terms becomes implicit. As shown in (Dumoulin et al., 2017; Donahue et al., 2017), this can cause original samples to differ substantially from their corresponding reconstructions. In order to address this drawback, we use a similar strategy as (Pu et al., 2018; Li et al., 2017), and add the reconstruction terms:

$$\mathbb{E}_{q_\psi(\mathbf{e}|\mathbf{x})q_\psi(\mathbf{e}|\mathbf{x})p^u(\mathbf{x})} \log(p_\theta(\mathbf{x}|\mathbf{z}, \mathbf{e})) + \mathbb{E}_{q_\psi(\mathbf{z}|\mathbf{x})p^r(\mathbf{x})} \log(p_\theta(\mathbf{x}|\mathbf{z}, \mathbf{e}^r)) +$$

$$\mathbb{E}_{p_\theta(\mathbf{x}|\mathbf{z},\mathbf{e})p(\mathbf{z})p(\mathbf{e})}[\log(q_\psi(\mathbf{z}|\mathbf{x})) + \log(q_\psi(\mathbf{e}|\mathbf{x}))] + \mathbb{E}_{p_\theta(\mathbf{x}|\mathbf{z},\mathbf{e}^r)p(\mathbf{z})} \log(q_\psi(\mathbf{z}|\mathbf{x})) \quad (10)$$

into the learning objective, minimizing them together with Eq. (8). In preliminary experiments, we found that the explicit addition of these reconstructions terms during learning is important to achieve low reconstruction errors and increase stability during the adversarial training procedure. Figure 5 in the Appendix illustrates all the different losses minimized during learning in sRBD-VAE.

## 5 EXPERIMENTS

### 5.1 DATASETS

To validate our approach and to compare to existing work, we consider two different reference-based disentangling problems.

**Digit style disentangling:** The goal is to model style variations from hand-written digits. Concretely, we consider the digit style as a set of three different properties: scale, width and color. In order to address this task from a reference-based perspective, we consider half of the original training images in the MNIST dataset (LeCun et al., 1998) as our reference distribution (30k examples). The unlabelled set is synthetically generated by applying different transformations over the remaining half of images: (1) Simulation of stroke widths by using a dilation with a given filter size; (2) Digit colorization by multiplying the RGB components of each pixel by a 3D RGB color vector; (3) Size variations by down-scaling the image by a given factor and applying zero-padding to recover the original resolution. We randomly transform each image twice to obtain a total of 60k unsuperivsed images. More details can be found in Appendix C.

**Facial expression disentangling:** We consider disentangling of facial expressions by using a reference set of neutral faces. For the unlabelled dataset we use a subset of the AffectNet dataset (Mollahosseini et al., 2017), which contains a large quantity of facial images. This dataset is especially challenging since images were collected "in the wild" and exhibit a large variety of natural expressions. A subset of the images are annotated according to different basic facial expressions: *happiness, sadness, surprise, fear, disgust, anger*, and *contempt*. We use these labels for quantitative evaluation. Given that we found that many neutral images in the original database were not correctly annotated, we collected a separate reference set, see Appendix C for details. The unlabelled and reference sets consist of 150k and 10k images, respectively.

### 5.2 EXPERIMENTAL SETUP

**Baselines:** We compare the two different variants of our proposed method: Rb-VAE, trained using the standard variational objective (Section 4.2), and sRb-VAE, learned by minimizing the symmetric KL divergence (Section 4.3). To demonstrate the advantages of exploiting the weak-supervision provided by reference images, we compare both methods with various state-of-the-art unsupervised approaches based on the VAE framework: $\beta$-VAE (Higgins et al., 2017), $\beta$-TCVAE (Chen et al., 2018) sVAE (Pu et al., 2018), DIP-VAE-I and DIP-VAE-II (Kumar et al., 2018). Note that $\beta$-VAE DIP-VAE and have $\beta$-TCVAE been specifically proposed for learning disentangled representations, showing better performance than other unsupervised methods such as InfoGAN (Chen et al., 2016). On the other hand, sVAE is trained using a similar variational objective as sRb-VAE, and can therefore be considered an unsupervised version of our method. We also use standard VAE (Kingma &

| | AffectNet | | | | | | | | MNIST | | | | | |
|---|---|---|---|---|---|---|---|---|---|---|---|---|---|---|
| | Happ | Sad | Sur | Fear | Disg | Ang | Compt | Avg. | R | G | B | Scale | Width | Avg. |
| VAE | .554 | .279 | .383 | .357 | .256 | .415 | .439 | .383 | .099 | .104 | .101 | [.034] | **.085** | .085 |
| DIP-VAE-I | .561 | .269 | [.401] | .367 | .258 | .397 | .463 | .388 | [.055] | **.064** | .063 | .038 | .100 | **.064** |
| DIP-VAE-II | .548 | .245 | [.401] | [.389] | .268 | .391 | .463 | .386 | .077 | .069 | .076 | **.035** | .098 | .071 |
| $\beta$ VAE | .581 | .283 | .373 | .323 | .250 | .415 | .467 | .384 | .093 | .099 | .094 | .039 | .089 | .083 |
| sVAE | **.583** | .251 | .389 | .349 | .260 | .391 | .469 | .384 | .094 | .092 | .084 | .036 | .104 | .082 |
| $\beta$-TCVAE | .563 | .277 | **.393** | .349 | .256 | [.427] | .467 | .390 | .098 | .100 | .099 | [.034] | [.084] | .083 |
| [Mathieu et. al, 2016] | .567 | .388 | .312 | .330 | .295 | .353 | [.512] | **.395** | .116 | .116 | .114 | .039 | .104 | .098 |
| RBD-VAE | .536 | **.393** | .379 | .311 | **.320** | .383 | .421 | 392 | .065 | .069 | **.062** | .061 | .095 | .070 |
| sRBD-VAE | [.587] | [.405] | .387 | .327 | [.344] | **.425** | **.483** | [.422] | **.057** | [.053] | [.055] | .038 | .095 | [.060] |

Table 1: Prediction of target factors from learned representations. For Rb-VAE and sRb-VAE only the latent variables **e** are used as feature vectors. All the latent variables are employed for unsupervised models. We report accuracy and Mean Absolute Error as evaluation metrics in the AffectNet and MNIST datasets respectively. Two best methods shown in bold, best result in brackets.

Welling, 2014) as an additional baseline. Finally, in order to evaluate an alternative approach able to exploit the reference-set, we have implemented (Mathieu et al., 2016) adapting it to our context. Note that the amount of supervision used in the cited method is larger than the available in reference-based disentangling. Concretely, we do not have information about unlabelled samples sharing the same target factors (e.g the same expression). In contrast, this information is only available for reference images. This fact renders the original learning algorithm proposed in (Mathieu et al., 2016) inapplicable. However, we have modified it in order to use only pairing information from reference images by removing the reconstruction losses assuming known pairs for unlabelled samples.

**Implementation details:** The different components of our method are implemented as deep neural networks. For this purpose, we have used an architecture based on the main building blocks employed by (Karras et al., 2018). More concretely, generator networks are implemented as a sequence of convolutions, Leaky-ReLU non-linearities, and nearest-neighbour up-sampling operations. Encoders and discriminators follow a similar architecture, using average pooling for down-sampling. Channel normalization is used for generators and encoders. See Appendix D for more details on the precise architecture. Our code will be made available online upon publication. For a fair comparison, we have developed our own implementation for all the evaluated methods in order to use the same network architectures and hyper-parameters. During optimization, SGD is employed using the Adam optimizer (Kingma & Ba, 2015) (with $\alpha = 10^{-4}, \beta_1 = 0.5, \beta_2 = 0.99, \epsilon = 10^{-8}$), and a batch size of 36 images. For the MNIST and AffectNet databases, the models are learned for 30 and 20 epochs respectively. The number of latent variables for the encoders has been set to 32 for all the experiments and models. The $\lambda$ parameter in the Laplace distribution is set to 0.01.

### 5.3 Quantitative evaluation: Feature Learning

A common strategy to evaluate the quality of learned representations is to measure the amount of information that they convey about the underlying generative factors (Eastwood & Williams, 2018). In the context of reference-based disentangling, we are interested in modelling the target factors that are constant in the reference distribution. In this experiment, we use this observation in order to quantitatively evaluate our method. For this purpose, we use the learned representations as feature vectors and train a low-capacity model estimating the target factors involved in each problem. Concretely, in the MNIST dataset we employ a set of linear-regressors predicting the scale, width and color parameters for each digit. To predict the different expression classes in the AffectNet dataset, we use a linear classifier. For evaluation, we split each dataset in three subsets. The first is used to learn each generative model. Then, the second is used for training the regressors or classifier. The third is used to evaluate the predictions in terms of the mean absolute error and per-class accuracy for the MNIST and AffectNet datasets, respectively. In MNIST, the second and third subset have been randomly generated from the original MNIST test set using the procedure described in Section 5.1 (5k images each). For AffectNet, we randomly select 500 images for each of the seven expressions from the original dataset, yielding 3,500 images per fold.

Table 1 shows the results obtained by the unsupervised models considered [1] and the proposed Rb-VAE and sRb-VAE. For Rb-VAE and sRb-VAE, we used the inferred latent variables **e** as features

---

[1]For DIP-VAE-I, DIP-VAE-II, $\beta$-VAE and $\beta$-TCVAE we tested different regularization parameters in the range $[1, 50]$, and report the best results.

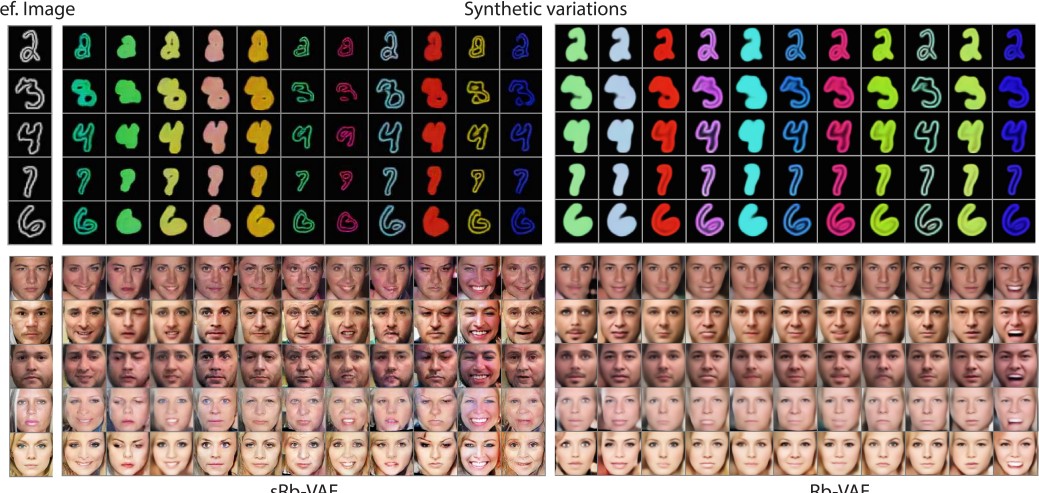

Figure 3: Conditional image synthesis for MNIST (top) and AffectNet (bottom) using sRb-VAE and Rb-VAE. Within each column images are generated using the same random target factors **e**.

since the distribution over these variables is expected to encode the information regarding the target factors. For the unsupervised models we use all the latent variables. By analyzing the results, we obtain the following conclusions. First, the unsupervised approach DIP-VAE-I achieves better average results than Rb-VAE for MNIST. Moreover, for facial expression recognition, $\beta$-TCVAE achieves comparable or better performance in several cases. This may seem counter-intuitive because, unlike Rb-VAE, DIP-VAE-I is trained without the weak-supervision provided by reference images. However, it confirms our hypothesis that the learning objective of Rb-VAE does not explicitly encourage the disentanglement between target and common factors. In contrast, we can see that sRb-VAE obtains comparable or better results than rest of the methods in most cases. Moreover, it achieves the best average performance in both datasets. This demonstrates that the information provided by the reference distribution is effectively exploited by the symmetric KL objective used to train sRb-VAE. In order to further validate the ability of the proposed method to disentangle the target factors, we have followed the same evaluation protocol for Rb-VAE and sRb-VAE but considering the latent variables **z** as features. The average performance obtained by Rb-VAE is *.349* and *.195* for AffectNet and MNIST respectively. On the other hand, sRb-VAE achieves *.335* and *.189*. Note that for both methods these results are significantly worse compared to using **e** as a representation in Table 1. This suggests that latent variables **z** are mainly modelling the common factors between reference and unlabelled images. The qualitative results presented in the next section confirm this. To conclude, note that sRBD-VAE also obtains better performance than (Mathieu et al., 2016) in both data-sets. So even though this method also uses reference images during training, our results confirms that sRBD-VAE better exploits the weak-supervision provided in reference-based disentangling.

## 5.4 QUALITATIVE EVALUATION

In contrast to unsupervised approaches, reference images can be used by our model in order to explicitly encode the target and common factors into two different subsets of latent variables. This directly enables tasks such as conditional image synthesis or attribute transfer. In this section, we illustrate the potential applications of our proposed model in this type of applications.

**Conditional image synthesis.** The goal is to transform real images by modifying only the target factors **e**. For instance, given a face of an individual, we aim to generate images of the same subject exhibiting different facial expressions. For this purpose, we use our model in order to infer the common factors **z**. Then, we sample a vector $\mathbf{e} \sim \mathcal{N}(\mathbf{0}, \mathbf{1})$ and use the generator network to obtain a new image from **e** and **z**. In Fig. 3 we show examples of samples generated by Rb-VAE and sRb-VAE following this procedure. As we can observe, sRb-VAE generates more convincing results than its non-symmetric counterpart. In the AffectNet database, the amount of variability in Rb-VAE

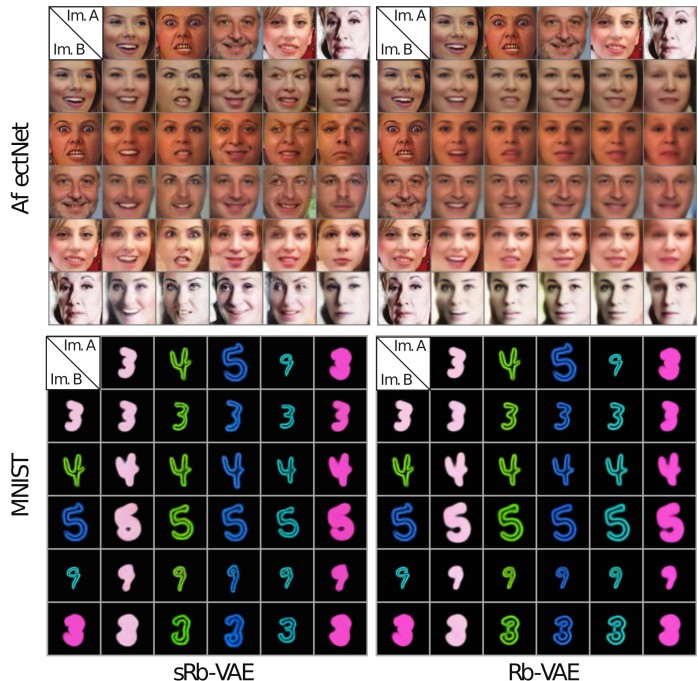

Figure 4: Transferring target factors **e** from image A to an image B on the AffectNet (expression) and MNIST (style) using sRb-VAE and Rb-VAE.

samples is quite low. In contrast, sRb-VAE is able to generate more diverse expressions related with eyes, mouth and eyebrows movements. Looking at the MNIST samples, we can draw similar conclusions. Whereas both methods generate transformations related with the digit color, Rb-VAE does not model scale variations in **e**, while sRb-VAE does. This observation is coherent with results reported in Table 1, where Rb-VAE offers a poor estimation of the scale.

**Visual attribute transfer.** Here we transfer target factors **e** between a pair of images A and B. For example, given two samples from the MNIST dataset, the goal is to generate a new image with the number in A modified with the style in B. Using our model, this can be easily achieved by synthesizing a new image from latent variables **e** and **z** inferred from A and B respectively. Figure 4 shows images generated by sRb-VAE and Rb-VAE in this scenario. In this case, we can draw similar conclusions than the previous experiment. Rb-VAE is not able to swap target factors related with the digit scale in the MNIST dataset, unlike sRb-VAE which better model this variation factor. On the AffectNet images, both methods are able to keep most of the information regarding the identity of the subject, but again Rb-VAE leads to weaker expression changes than sRb-VAE.

These qualitative results demonstrate that the standard variational objective of VAE is sub-optimal to train our model, and that the symmetric KL divergence objective used in sRb-VAE allows to better disentangle the common and target factors.

## 6 CONCLUSIONS

In this paper we have introduced reference-based disentangling, a novel weakly supervised learning problem, and proposed reference-based variational autoencoder models to address it. We have shown that the standard variational learning objective used to train VAE can lead to degenerate solutions when it is applied in our setting, and proposed an alternative training strategy that exploits adversarial learning. Comparing the proposed model with previous state-of-the-art unsupervised approaches, we have demonstrated its ability to learn disentangled representations from minimal supervision, and its potential application to tasks such as feature learning, conditional image generation, and attribute transfer.

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

## APPENDIX A    MATHEMATICAL DERIVATIONS

**Equivalence between $\mathbb{KL}(q_\psi(\mathbf{z}, \mathbf{e}, \mathbf{x}, y) \parallel p_\theta(\mathbf{x}, \mathbf{z}, \mathbf{e}, y))$ and Eq. (3):**

$$\sum_{y \in [0,1]} \int_{x,e,z} q_\psi(\mathbf{e}|\mathbf{x}, y) q_\psi(\mathbf{z}|\mathbf{x}) p(\mathbf{x}|y) p(y) \log \left( \frac{q_\psi(\mathbf{z}, \mathbf{e}|\mathbf{x}, y) p(\mathbf{x}|y) p(y)}{p_\theta(\mathbf{x}|\mathbf{e}, \mathbf{z}) p(\mathbf{z}) p(\mathbf{e}|y) p(y)} \right) d\mathbf{x} d\mathbf{z} d\mathbf{e} \tag{11}$$

$$= \frac{1}{2} \int_{x,e,z} q_\psi(\mathbf{e}|\mathbf{x}) q_\psi(\mathbf{z}|\mathbf{x}) p^u(\mathbf{x}) \log \left( \frac{q_\psi(\mathbf{e}|\mathbf{x}) q_\psi(\mathbf{z}|\mathbf{x}) p^u(\mathbf{x})}{p_\theta(\mathbf{x}|\mathbf{e}, \mathbf{z}) p(\mathbf{z}) p(\mathbf{e})} \right) d\mathbf{x} d\mathbf{z} d\mathbf{e}$$

$$+ \frac{1}{2} \int_{x,z} q_\psi(\mathbf{z}|\mathbf{x}) p^r(\mathbf{x}) \log \left( \frac{q_\psi(\mathbf{z}|\mathbf{x}) p^r(\mathbf{x})}{p_\theta(\mathbf{x}|\mathbf{e}^r, \mathbf{z}) p(\mathbf{z})} \right) d\mathbf{x} d\mathbf{z} \tag{12}$$

$$= \frac{1}{2} \mathbb{E}_{p^u(\mathbf{x})} \mathbb{E}_{q_\psi(\mathbf{e}|\mathbf{x}) q_\psi(\mathbf{z}|\mathbf{x})} \left[ \log \left( \frac{q_\psi(\mathbf{e}|\mathbf{x}) q_\psi(\mathbf{z}|\mathbf{x})}{p(\mathbf{z}) p(\mathbf{e})} \right) - \log(p_\theta(\mathbf{x}|\mathbf{e}, \mathbf{z})) \right] - H^u(\mathbf{x})$$

$$+ \frac{1}{2} \mathbb{E}_{p^r(\mathbf{x})} \mathbb{E}_{q_\psi(\mathbf{z}|\mathbf{x})} \left[ \log \left( \frac{q_\psi(\mathbf{z}|\mathbf{x})}{p(\mathbf{z})} \right) - \log(p_\theta(\mathbf{x}|\mathbf{e}^r, \mathbf{z})) \right] - H^r(\mathbf{x}) \tag{13}$$

$$= \frac{1}{2} \mathbb{E}_{p^u(\mathbf{x})} \left[ \mathbb{KL}(q_\psi(\mathbf{z}|\mathbf{x}) q_\psi(\mathbf{e}|\mathbf{x}) \parallel p(\mathbf{z}) p(\mathbf{e})) - \mathbb{E}_{q_\psi(\mathbf{z}|\mathbf{x}) q_\psi(\mathbf{e}|\mathbf{x})} \log(p_\theta(\mathbf{x}|\mathbf{z}, \mathbf{e})) \right]$$

$$+ \frac{1}{2} \mathbb{E}_{p^r(\mathbf{x})} \left[ \mathbb{KL}(q_\psi(\mathbf{z}|\mathbf{x}) \parallel p(\mathbf{z})) - \mathbb{E}_{q_\psi(\mathbf{z}|\mathbf{x})} \log(p_\theta(\mathbf{x}|\mathbf{z}, \mathbf{e}^r)) \right] - H^r(\mathbf{x}) - H^u(\mathbf{x})$$

We denote $H^r(X)$ and $H^u(X)$ as the entropy of the reference and unlabelled distributions $p^r(\mathbf{x})$ and $p^u(\mathbf{x})$ respectively. Note that they can be ignored during the minimization since are constant w.r.t. parameters $\theta$ and $\psi$. For the second equality, we have used the definitions $p(\mathbf{x}|y = 0) = p^u(\mathbf{x})$, $p(\mathbf{x}|y = 1) = p^r(\mathbf{x})$ and assumed $p(y = 0) = p(y = 1) = \frac{1}{2}$. Moreover, we have exploited the fact that $q_\psi(\mathbf{e}|\mathbf{x}, y = 1)$ and $p(\mathbf{e}|y = 1)$ are defined as delta functions and, therefore, $\mathbb{E}_{p(\mathbf{e}|y=1)} log(\frac{p(\mathbf{e}|y=1)}{q_\psi(\mathbf{e}|y=1)}) = 0$. We denote $p(\mathbf{e}|y = 0) = p(\mathbf{e})$ and $q_\psi(\mathbf{e}|\mathbf{x}, y = 0) = q_\psi(\mathbf{e}|\mathbf{x})$ for the sake of brevity.

**Equivalence between $\mathbb{KL}(p_\theta(\mathbf{x}, \mathbf{z}, \mathbf{e}, y) \parallel q_\psi(\mathbf{z}, \mathbf{e}, \mathbf{x}, y))$ and the expression in Eq. (5)**

$$\sum_{y \in [0,1]} \int_{x,e,z} p_\theta(\mathbf{x}|\mathbf{e}, \mathbf{z}) p(\mathbf{z}) p(\mathbf{e}|y) p(y) \log \left( \frac{p_\theta(\mathbf{x}|\mathbf{e}, \mathbf{z}) p(\mathbf{z}) p(\mathbf{e}|y) p(y)}{q_\psi(\mathbf{z}, \mathbf{e}, \mathbf{x}, y) p(\mathbf{x}|y) p(y)} \right) d\mathbf{x} d\mathbf{z} d\mathbf{e} \tag{14}$$

$$= \frac{1}{2} \int_{x,e,z} p_\theta(\mathbf{x}|\mathbf{e}, \mathbf{z}) p(\mathbf{z}) p(\mathbf{e}) \log \left( \frac{p_\theta(\mathbf{x}|\mathbf{e}, \mathbf{z}) p(\mathbf{z}) p(\mathbf{e})}{q_\psi(\mathbf{e}|\mathbf{x}) q_\psi(\mathbf{z}|\mathbf{x}) p^U(\mathbf{x})} \right) d\mathbf{x} d\mathbf{z} d\mathbf{e}$$

$$+ \frac{1}{2} \int_{x,z} p_\theta(\mathbf{x}|\mathbf{e}^r, \mathbf{z}) p(\mathbf{z}) \log \left( \frac{p_\theta(\mathbf{x}|\mathbf{e}^r, \mathbf{z}) p(\mathbf{z})}{q_\psi(\mathbf{z}|\mathbf{x}) p^u(\mathbf{x})} \right) d\mathbf{x} d\mathbf{z} d\mathbf{e} \tag{15}$$

$$= \frac{1}{2} \mathbb{E}_{p(\mathbf{z}) p(\mathbf{e})} \mathbb{E}_{p_\theta(\mathbf{x}|\mathbf{e}, \mathbf{z})} \left[ \log \left( \frac{p_\theta(\mathbf{x}|\mathbf{e}, \mathbf{z})}{p(\mathbf{x})^u} \right) - \log(q_\psi(\mathbf{e}|\mathbf{x}) q_\psi(\mathbf{z}|\mathbf{x})) \right]$$

$$+ \frac{1}{2} \mathbb{E}_{p(\mathbf{z})} \mathbb{E}_{p_\theta(\mathbf{x}|\mathbf{e}^r, \mathbf{z})} \left[ \log \left( \frac{p_\theta(\mathbf{x}|\mathbf{e}^r, \mathbf{z})}{p(\mathbf{x})^r} \right) - \log(q_\psi(\mathbf{z}|\mathbf{x})) \right] - H(\mathbf{z}) - \frac{1}{2} H(\mathbf{e}) \tag{16}$$

$$= \frac{1}{2} \mathbb{E}_{p(\mathbf{z}) p(\mathbf{e})} \left[ \mathbb{KL}(p_\theta(\mathbf{x}|\mathbf{z}, \mathbf{e}) \parallel p^u(\mathbf{x})) - \mathbb{E}_{p_\theta(\mathbf{x}|\mathbf{z}, \mathbf{e})} [\log(q_\psi(\mathbf{z}|\mathbf{x})) + \log(q_\psi(\mathbf{e}|\mathbf{x}))] \right]$$

$$\frac{1}{2} \mathbb{E}_{p(\mathbf{z})} \left[ \mathbb{KL}(p_\theta(\mathbf{x}|\mathbf{z}, \mathbf{e}^r) \parallel p^r(\mathbf{x})) - \mathbb{E}_{p_\theta(\mathbf{x}|\mathbf{z}, \mathbf{e}^r)} \log(q_\psi(\mathbf{z}|\mathbf{x})) \right] - H(\mathbf{z}) - \frac{1}{2} H(\mathbf{e}) \tag{17}$$

We have used the same definitions and assumptions previously discussed. Moreover, we denote $H(\mathbf{z})$ and $H(\mathbf{e})$ as the entropy of the priors $p(\mathbf{z})$ and $p(\mathbf{e})$. Again, we can ignore these terms when we are optimizing w.r.t parameters $\psi$ and $\theta$.

**Equivalence between the minimization of the symmetric KL divergence in Eq. (4) and the expression in Eq. (6)**

$$\mathbb{KL}(q_\psi(\mathbf{z}, \mathbf{e}, \mathbf{x}, y) \parallel p_\theta(\mathbf{x}, \mathbf{z}, \mathbf{e}, y)) + \mathbb{KL}(p_\theta(\mathbf{x}, \mathbf{z}, \mathbf{e}, y) \parallel q_\psi(\mathbf{z}, \mathbf{e}, \mathbf{x}, y)) = \tag{18}$$

$$= \mathbb{E}_{q_\psi(\mathbf{e}|\mathbf{x},y)q_\psi(\mathbf{z}|\mathbf{x})p(\mathbf{x}|y)p(y)} \log\left( \frac{q_\psi(\mathbf{e}|\mathbf{x}, y)q_\psi(\mathbf{z}|\mathbf{x})p(\mathbf{x}|y)p(y)}{p_\theta(\mathbf{x}|\mathbf{e}, \mathbf{z})p(\mathbf{z})p(\mathbf{e}|y)p(y)} \right)$$

$$+ \mathbb{E}_{p_\theta(\mathbf{x}|\mathbf{e},\mathbf{z})p(\mathbf{z})p(\mathbf{e}|y)p(y)} \log\left( \frac{p_\theta(\mathbf{x}|\mathbf{e}, \mathbf{z})p(\mathbf{z})p(\mathbf{e}|y)p(y)}{q_\psi(\mathbf{e}|\mathbf{x}, y)q_\psi(\mathbf{z}|\mathbf{x})p(\mathbf{x}|y)p(y)} \right) \tag{19}$$

$$= \frac{1}{2}\left[ \mathbb{E}_{q_\psi(\mathbf{e},\mathbf{z}|\mathbf{x})p^u(\mathbf{x})} \log\left( \frac{q_\psi(\mathbf{e}, \mathbf{z}|\mathbf{x})p^u(\mathbf{x})}{p_\theta(\mathbf{x}|\mathbf{e}, \mathbf{z})p(\mathbf{z})p(\mathbf{e})} \right) + \mathbb{E}_{q_\psi(\mathbf{z}|\mathbf{x})p^r(\mathbf{x})} \log\left( \frac{q_\psi(\mathbf{z}|\mathbf{x})p(\mathbf{x})^r}{p_\theta(\mathbf{x}|\mathbf{e}^r, \mathbf{z})p(\mathbf{z})} \right) \right.$$

$$\left. + \mathbb{E}_{p_\theta(\mathbf{x}|\mathbf{e},\mathbf{z})p(\mathbf{z})p(\mathbf{e})} \log\left( \frac{p_\theta(\mathbf{x}|\mathbf{e}, \mathbf{z})p(\mathbf{z})p(\mathbf{e})}{q_\psi(\mathbf{e}, \mathbf{z}|\mathbf{x})p^u(\mathbf{x})} \right) + \mathbb{E}_{p_\theta(\mathbf{x}|\mathbf{e}^r,\mathbf{z})p(\mathbf{z})} \log\left( \frac{p_\theta(\mathbf{x}|\mathbf{e}^r, \mathbf{z})p(\mathbf{z}))}{q_\psi(\mathbf{z}|\mathbf{x})p^r(\mathbf{x})} \right) \right] \tag{20}$$

## APPENDIX B  PSEUDO-CODE FOR ADVERSARIAL LEARNING PROCEDURE

Alg. 1 shows pseudo-code for the adversarial learning algorithm described in Sec. 4.3 of the paper. Fig. 5 illustrates all the different losses minimized during learning in sRBD-VAE.

---

**Algorithm 1** sRb-VAE Advesarial Learning (Batch processing during SGD)

---

1: /*** Gradient $\phi$ ***/
2: Sample $\{\mathbf{x}_1, ..., \mathbf{x}_M\}$ from $p^u(\mathbf{x})$
3: Sample $\{\mathbf{x}_1^r, ..., \mathbf{x}_M^r\}$ from $p^r(\mathbf{x})$
4: Sample $\{\mathbf{e}_1, ..., \mathbf{e}_M\}$ using $q_\phi(\mathbf{e}|\mathbf{x})$
5: Sample $\{\mathbf{z}_1, ..., \mathbf{z}_M\}$ using $q_\phi(\mathbf{z}|\mathbf{x})$
6: Sample $\{\mathbf{z}_1^r, ..., \mathbf{z}_M^r\}$ using $q_\phi(\mathbf{z}|\mathbf{x}^r)$
7: Compute gradient of Eq. (8) w.r.t $\psi$ using the reparametrization trick for stochastic variables $\mathbf{z}$, $\mathbf{e}$ and $\mathbf{z}^r$:

$$g_\phi \leftarrow \nabla_\phi \frac{1}{m}\left[ \sum_m d_\xi(\mathbf{x}_m, \mathbf{z}_m, \mathbf{e}_m) \right.$$
$$\left. + d_\gamma(\mathbf{x}_m^r, \mathbf{z}_m^r) \right]$$

8: /*** Gradient $\theta$ ***/
9: Sample $\{\hat{\mathbf{e}}_1, ..., \hat{\mathbf{e}}_M\}$ from $p(\mathbf{e})$
10: Sample $\{\hat{\mathbf{z}}_1, ..., \hat{\mathbf{z}}_M\}$ from $p(\mathbf{z})$
11: Sample $\{\hat{\mathbf{z}}_1^r, ..., \hat{\mathbf{z}}_M^r\}$ from $p(\mathbf{z})$
12: Sample $\{\hat{\mathbf{x}}_1, ..., \hat{\mathbf{x}}_M\}$ using $p_\theta(\mathbf{x}|\hat{\mathbf{z}}, \mathbf{e})$
13: Sample $\{\hat{\mathbf{x}}_1^r, ..., \hat{\mathbf{x}}_M^r\}$ using $p_\theta(\mathbf{x}|\hat{\mathbf{z}}, \mathbf{e}^r)$
14: Compute gradient of Eq. (8) w.r.t $\theta$ using the reparametrization trick for stochastic variables $\hat{\mathbf{x}}$ and $\hat{\mathbf{x}}^r$:

$$g_\theta \leftarrow \nabla_\theta \frac{1}{m}\left[ \sum_m d_\xi(\hat{\mathbf{x}}_m, \hat{\mathbf{z}}_m, \hat{\mathbf{e}}_m) \right.$$
$$\left. + d_\gamma(\hat{\mathbf{x}}_m^r, \hat{\mathbf{z}}_m^r) \right]$$

15: /*** Gradient $\xi$ ***/
16: Compute gradient of discriminator function (Eq. (9)) w.r.t $\xi$:

$$g_\xi \leftarrow \nabla_\xi \frac{1}{2m} \sum_m \left[ \log(\sigma(d_\xi(\mathbf{x}_m, \mathbf{z}_m, \mathbf{e}_m))+ \right.$$
$$\left. \log(1 - \sigma(d_\xi(\hat{\mathbf{x}}_m, \hat{\mathbf{z}}_m, \hat{\mathbf{e}}_m)) \right]$$

17: /*** Gradient $\gamma$ ***/
18: Compute gradient of discriminator function (Eq. (9)) w.r.t $\gamma$:

$$g_\gamma \leftarrow \nabla_\gamma \frac{1}{2m} \sum_m \left[ \log(\sigma(d_\gamma(\mathbf{x}_m^r, \mathbf{z}_m^r))+ \right.$$
$$\left. \log(1 - \sigma(d_\gamma(\hat{\mathbf{x}}_m^r, \hat{\mathbf{z}}_m^r)) \right]$$

19: /*** Update Parameters ***/
20: Update parameters via SGD with learning rate $\lambda$:

$$\theta \leftarrow \theta + \lambda g_\theta$$
$$\psi \leftarrow \psi - \lambda g_\psi$$
$$\xi \leftarrow \xi + \lambda g_\xi$$
$$\gamma \leftarrow \gamma + \lambda g_\gamma$$

$$=0$$

---

## APPENDIX C  DATASETS

Examples of reference and unlabelled images for MNIST and AffectNet are shown in Fig. 6. Following, we provide more information about the used datasets.

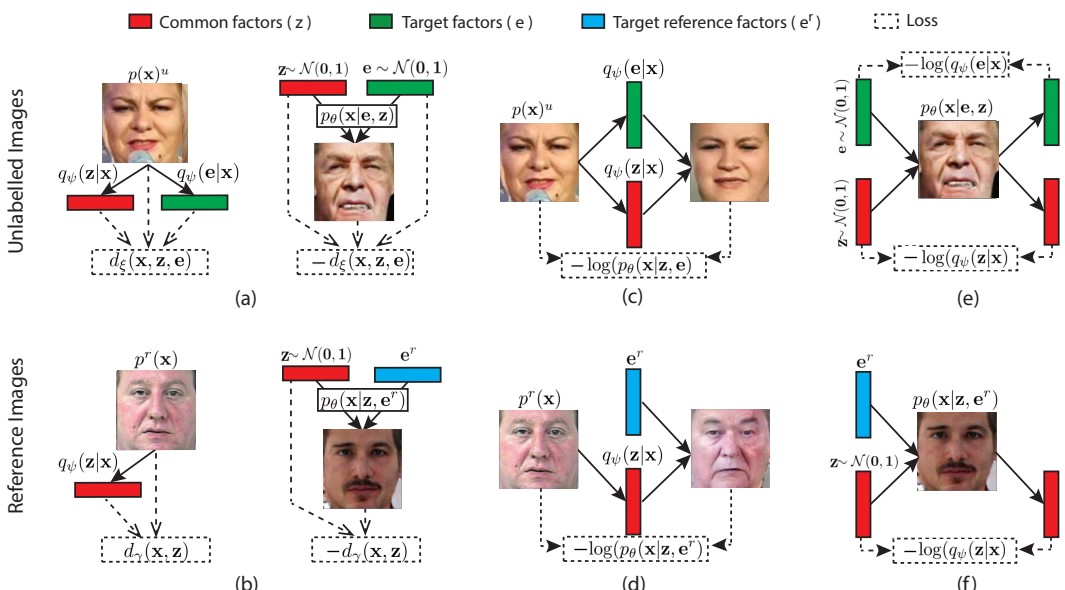

Figure 5: Illustration of the different losses minimized during learning in sRBD-VAE. (a) Model encoders and generator are trained to minimize the divergence between distributions $q_\psi(\mathbf{z}, \mathbf{e}|\mathbf{x})p^u(\mathbf{x})$ and $p_\theta(\mathbf{x}|\mathbf{e}, \mathbf{z})p(\mathbf{z})p(\mathbf{e})$. The discriminator $d_\varepsilon(\mathbf{x}, \mathbf{z}, \mathbf{e})$ measures the log-density ratio between both distributions. (b) A similar loss is minimized for reference images using an additional discriminator $d_\gamma(\mathbf{x}, \mathbf{z})$ (c,d) Encoder and generator aim to reconstruct unlabelled and reference images from their inferred latent variables. Note that for reference images, no inference over constant factors $\mathbf{e}$ is needed. (e,f) For unlabelled and reference images, sRDB-VAE also minimizes the reconstruction error over latent variables of images generated using $p(\mathbf{z})$, $p(\mathbf{e})$ and $\mathbf{e^r}$

## C.1 MNIST

We use slightly modified version of the MNIST images: the size is increased to $64 \times 64$ pixels and an edge detection procedure is applied to keep only the boundaries of the digit. We obtain the samples in the unlabelled dataset by applying the following transformations over the MNIST images:

1. **Width**: Generate a random integer in the range $\{1, \ldots, 10\}$ using a uniform distribution. Apply a dilation operation over the image using a squared kernel with pixel-size equal to the generated number.

2. **Color**: Generate a random 3D vector $c \in [0, 1]^3$ using a uniform distribution. Normalize the resulting vector as $\hat{c} = c/||c||_1$. Multiply the RGB components of all the pixels in the image by $\hat{c}$.

3. **Size**: Generate a random number in the range $[0.5, 1]$ using a uniform distribution. Downscale the image by a factor equal to the generated number. Apply zero-padding to the resulting image in order to recover the original resolution.

The data and code to generate it will be made available online upon publication.

## C.2 AFFECTNET

**Reference set collection.** We collected a reference set of face images with neutral expression. We applied specific queries in order to obtain a large amount of faces from image search engines. Then, five different annotators filtered them in order to keep only images showing a neutral expression. This was motivated because we found that many neutral images in the AffectNet dataset Mollahosseini et al. (2017) were not accurate. As detailed in the original paper, each image was only labelled by a single subject. As a consequence, the agreement between annotators was shown to be low for neutral images. In contrast, in our reference-set, each image was annotated in terms of "neutral" / "non-neutral" by two different annotators. In order to ensure a higher label quality compared to

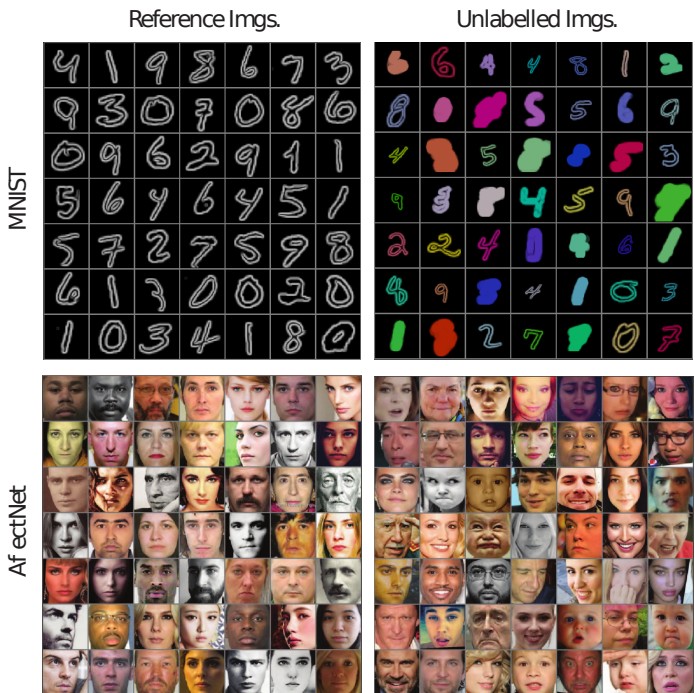

Figure 6: Examples of reference and unlabelled images used in our experiments. Extracted from MNIST (top) and AffectNet (bottom) databases.

the AffectNet, only the images where both annotators agreed were added to the reference-set. The collected dataset will be made available upon publication.

**Pre-processing.** In order to remove 2D affine transformations such as scaling or in-plane rotations, we apply an alignment process to the face images. We localize facial landmarks using (Xiong & De la Torre, 2013). Then, we apply Procrustes analysis in order to find an affine transformation aligning the detected landmarks with a mean shape. Finally, we apply the transformation to the image and crop it. The resulting image is then re-sized to a resolution of $96 \times 96$ pixels.

## APPENDIX D   NETWORK ARCHITECTURES

Fig. 7 illustrates the network architectures used in our experiments. CN refers to pixel-wise normalization as described in (Karras et al., 2018). FC defines a fully-connected layer. For Leaky ReLU non-linearities, we have used an slope of $0.2$. Given that we normalize the images in the range $[-1, 1]$, we use an hyperbolic tangent function as the last layer of the generator. For the discriminator $d_\gamma(\mathbf{x}, \mathbf{z})$, we use the same architecture showed for $d_\xi(\mathbf{x}, \mathbf{z}, \mathbf{e})$ but removing the input corresponding to $\mathbf{e}$. In preliminary experiments, we found that the discriminator in sRb-VAE can start to ignore the inputs corresponding to latent variables $\mathbf{e}$ and $\mathbf{z}$ while focusing only on real and generated images. In order to mitigate this problem during training, we found it effective to randomly set to zero the inputs corresponding to latent variables and images of the last fully-connected layer. In particular, we use a probability of $0.25$.

## APPENDIX E   ADDITIONAL RESULTS

Figures 8 and 9 show additional qualitative results for conditional image generation and visual attribute transfer, in the same spirit as the ones in 5.4. The additional results further support the conclusions drawn in the main paper. In Fig. 10, we also show additional images generated by sRB-VAE trained with the AffectNet dataset. Different from the previous cases, these images have been generated by injecting random noise over both latent variables $\mathbf{e}$ and $\mathbf{z}$. Note that different target factors $\mathbf{e}$ generate similar expressions in images generated from different common factors $\mathbf{z}$.

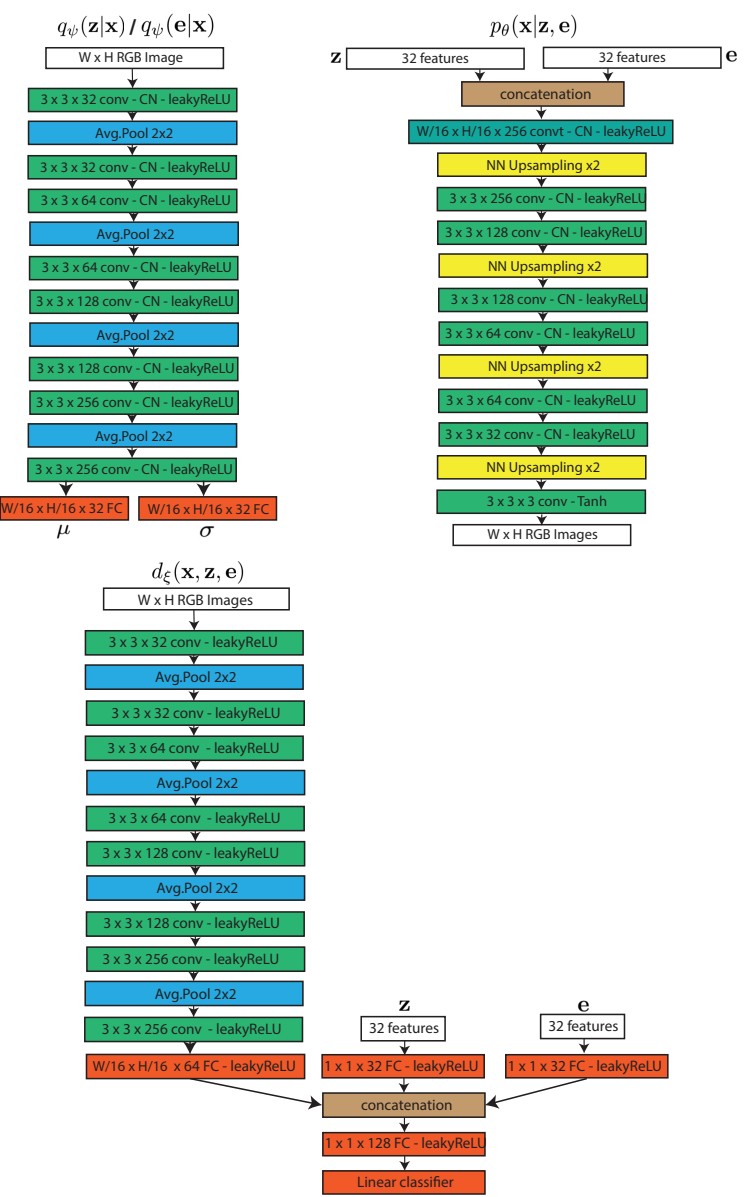

Figure 7: Network architectures used in our experiments

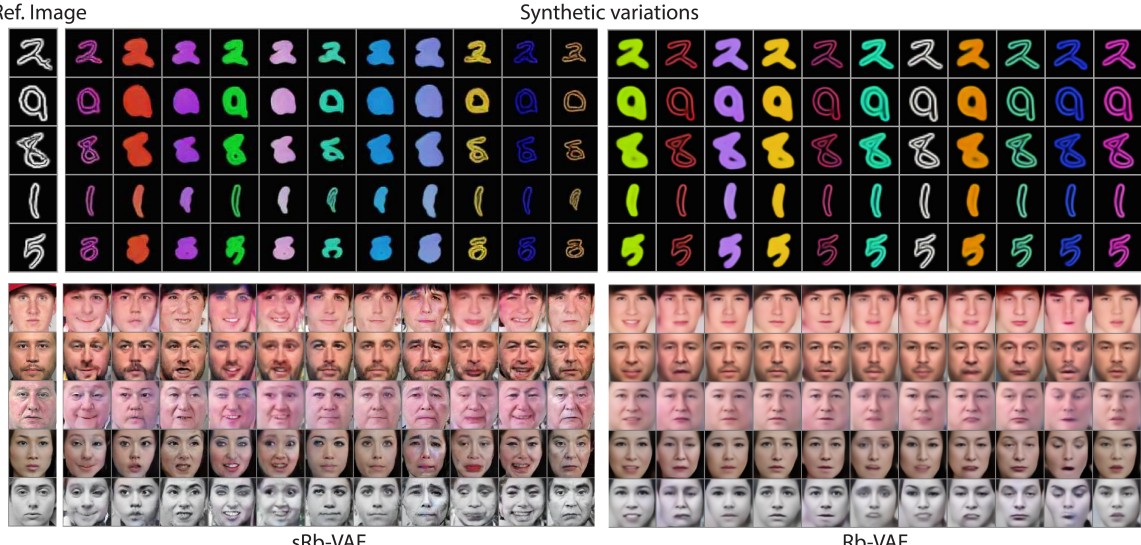

Figure 8: Qualitative results of sRb-VAE and Rb-VAE applied to conditional image generation. See Sec. (5.5) of the paper for details.

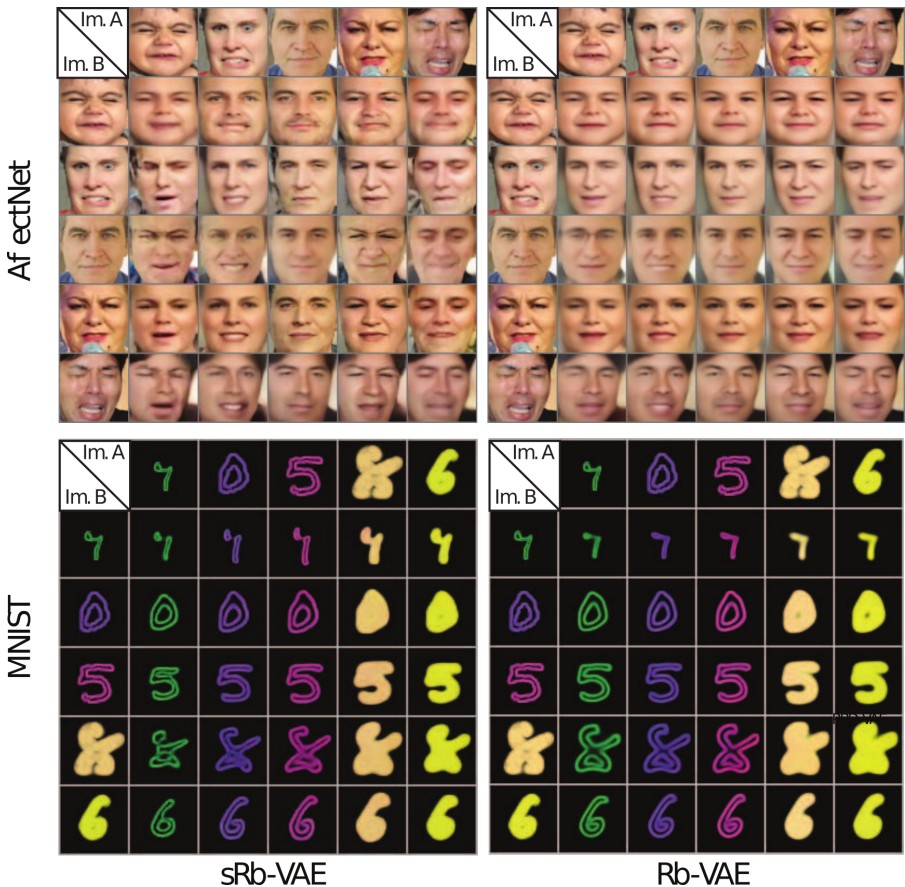

Figure 9: Qualitative results of sRb-VAE and Rb-VAE applied to visual attribute transfer. See Sec. (5.5) of the paper for details.

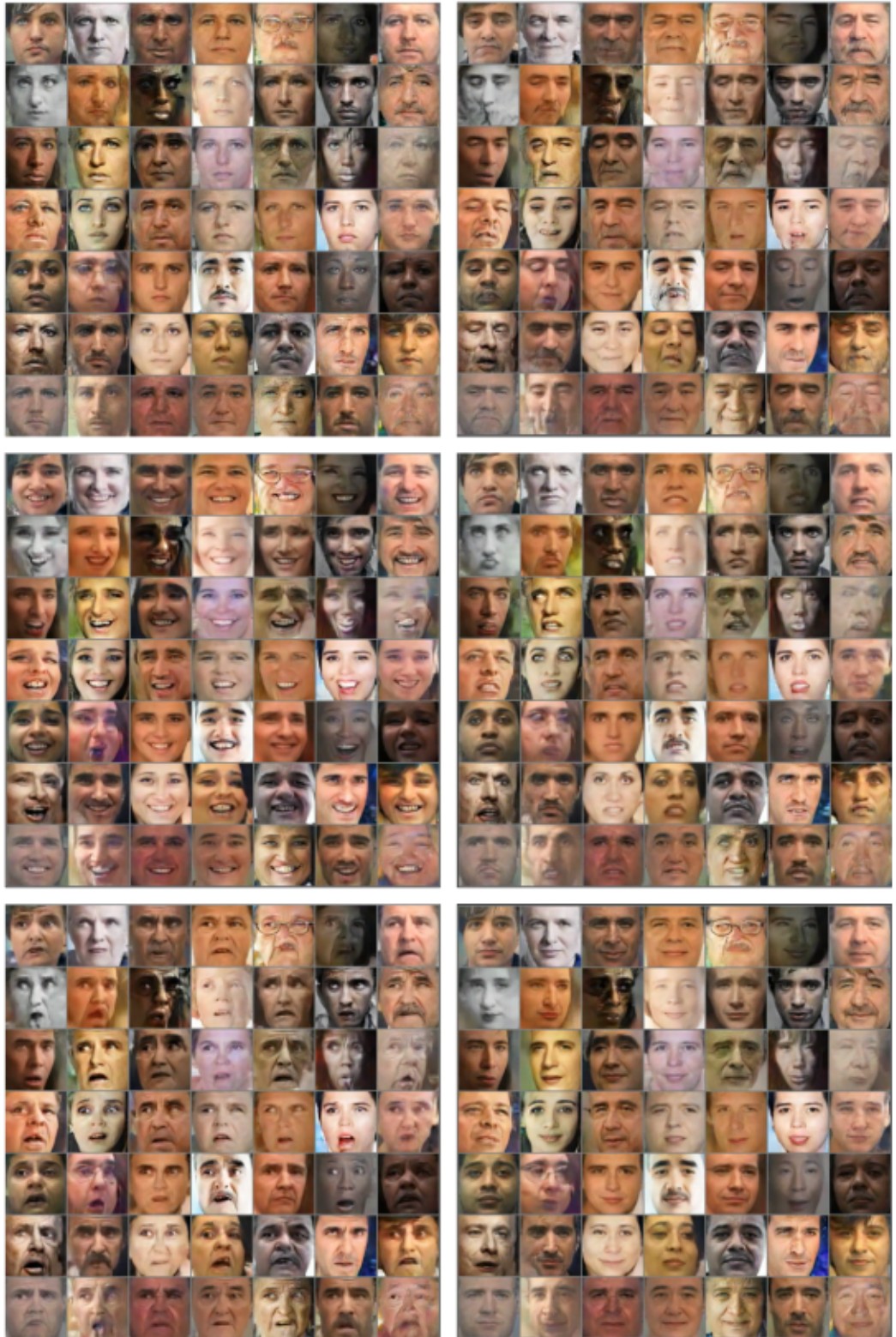

Figure 10: Images generated by sRB-VAE from random noise over latent variables **e** and **z**. Images in the same panel share the same target factors **e** (expression). Images sharing the same position in each panel are generated from the same common factors **z** (identity)

