# OpenReview forum: "Learning Disentangled Representations with Reference-Based Variational Autoencoders"
_ICLR.cc/2019/Conference_

### Official Review · AnonReviewer2 · 2018-11-01
**Results are promising, but missing comparison to an established method. And the loss seems more complicated than it need be.**

**Rating:** 6
**Confidence:** 4

**Review:**

Summary: Given two sets of data, where one is unlabelled and the other is a reference data set with a particular factor of variation that is fixed, the approach disentangles this factor of variation from the others. The approach uses a VAE whose latents are split into e that represents the factor of variation and z that represents the remaining factors. A symmetric KL loss that is approximated using the density-ratio trick is optimised for the learning, and the method is applied to MNIST digit style disentangling and AffectNet facial expression disentangling.

Pros:
- Clearly written
- Results look promising, both quantitative and qualitative.

Cons:
- Mathieu et al disentangle a specific factor from others without explicit labels but by drawing two images with the same value of the specified factor (i.e. drawing from the reference set) and also drawing a third image with a any value of the specified factor (i.e. drawing from the unlabelled set). Hence their approach is directly applicable to the problem at hand in the paper. Although Mathieu et al use digit/face identity as the shared factor, their method is directly applicable to the case where the shared factor is digit style/facial expression. Hence it appears to me that it should be compared against.
- missing reference - Bouchacourt - explicit labels aren’t given and data is grouped where each group shares a factor of var. But here the data is assumed to be partitioned into groups, so there is no equivalent to the unlablled set, hence difficult to compare against for the outlined tasks.
- Regarding comparison against unsupervised disentangling methods, there have been more recent approaches since betaVAE and DIP-VAE (e.g. FactorVAE (Kim et al) TCVAE (Chen et al)). It would be nice to compare against these methods, not only via predictive accuracy of target factors but also using disentangling metrics specified in these papers.

Other Qs/comments
- the KL terms in (5) are intractable due to the densities p^u(x) and p^r(x), hence two separate discriminators need to be used to approximate two separate density ratios, making the model rather large and complicated with many moving parts. What would happen if these KL terms in (5) are dropped and one simply uses SGVB to optimise the resulting loss without the need for discriminators? Usually discriminators tend to heavily underestimate density ratios (See e.g. Rosca et al), especially densities defined on high dimensions, so it might be best to avoid them whenever possible. The requirement of adding reconstruction terms to the loss in (10) is perhaps evidence of this, because these reconstruction terms are already present in the loss (3) & (5) that the discriminator should be approximating. So the necessity of extra regularisation of these reconstruction terms suggests that the discriminator is giving poor estimates of them. The reconstruction terms for z,e in (5) appear sufficient to force the model to use e (which is the motivation given in the paper for using the symmetric KL), akin to how InfoGAN forces the model to use the latents, so the necessity of the KL terms in (5) is questionable and appears to need further justification and/or ablation studies.
- (minor) why not learn the likelihood variance lambda?

************* Revision *************
I am convinced by the rebuttal of the authors, hence have modified my score accordingly.

---

> ### Author Response · Authors · 2018-11-12
> **Response to AnonReviewer2 [1/2]**
>
> We would like to thank the reviewer for his useful comments and suggestions. Detailed comments about specific concerns are addressed below.
>
>
>
> ***(R2.Q1) Comparison with [Mathieu et. al, 2016]
>
> We would like to clarify that the type of supervision used in [Mathieu et. al, 2016] is not equivalent to the one assumed in our problem. As discussed in R3.Q8, “the reference-based setting is different from the scenario  where information about samples sharing the same target factors is available. In particular, in our case we only know that reference images share the same label. In contrast, for the unlabelled distribution we do not have access to this information”. Note that this fact renders the original learning algorithm proposed in [Mathieu et. al, 2016] inapplicable in our context. The reason is that for any unlabelled image, we should be able to sample another image with the same generative factor (e.g. the same expression), given that we need to reconstruct them by swapping their latent representation e. Intuitively, this shows that reference-based disentangling is a more challenging problem than the one addressed in [Mathieu et. al, 2016]. The reason is that the amount of available supervision is lower (following the original paper nomenclature, only one type of “id” is labelled).
>
> Beyond the discussed difference, we agree with the reviewer that it is interesting to evaluate how the approach presented in [Mathieu et. al, 2016] behaves if only a single id is available (i.e, the reference label). For this reason, we have implemented this method using the same network architectures as in the rest of our models. Following the experimental evaluation described in Sec. 5.3, we have trained it by using the procedure suggested by the reviewer  (i.e, only pairs of reference images are used during training and no labels for unlabelled images are assumed). Note that this implies a modification over the original learning algorithm. We have added the results in Table 1. As can be seen, with the method of [Mathieu et. al, 2016] we obtain reasonable results in the AffectNet dataset. However, sRBD-VAE achieves better average accuracy. On the other hand, in the MNIST dataset, using the approach of [Mathieu et. al, 2016] obtains poor performance compared to most methods. These results confirm that our approach is better suited to exploit the weak-supervision provided by the reference set of images. We have added this evaluation into the revised paper (see baselines in Sec 5.2 and discussion of the results in Sec 5.3)
>
> It is also worth mentioning that an advantage of our model compared to [Mathieu et. al, 2016] is that we are able to naturally address conditional image generation (Sec 5.4) by sampling latent variables from p(e). Note that in [Mathieu et. al] this is not possible given that no prior over the target factors e is forced.
>
>
>
> ***(R2.Q2) “missing reference - Bouchacourt”
>
> We have added the reference in Related Work section.
>
>
>
> ***(R2.Q3) “ ...there have been more recent approaches since betaVAE and DIP-VAE (e.g. FactorVAE (Kim et al) TCVAE (Chen et al)). It would be nice to compare against these methods, not only via predictive accuracy of target factors but also using disentangling metrics specified in these papers…”
>
> We thank the reviewer for pointing out these recent unsupervised methods. We have added these references to the updated version of the paper. For the sake of completeness of our evaluation, we have implemented the method proposed by [Chen et. al,2018] (note that TCVAE and FactorVAE minimize the same objective function) and run the same experiments described in Sec. 5.3 of our paper.  We have added the quantitative results in Table 1 of the updated paper. Note that TCVAE obtain a similar average performance compared to other unsupervised approaches like bVAE or DIP-VAE. Moreover, the average results obtained by our method in both datasets are better. Therefore, our conclusions in this experiment remain unchanged.
>
> On the other hand, we would like to clarify that the metrics proposed in [Chen et. al, 2018, Kim et. al, 2018] are specifically designed for evaluating how a single dimension of the latent representation corresponds to a single ground-truth generative factor. As we have discussed in our response to Reviewer 1 (see Q1.R1), this is not the goal of our work and we believe that the one-to-one mapping assumption is not realistic when modelling high-level generative factors. For example, it is not reasonable to expect that a single dimension of the latent vector e can convey all the information about a complex generative factor such as the facial expression. Therefore, we think that the metrics proposed in the cited works are not appropriate in our context.

---

> ### Author Response · Authors · 2018-11-12
> **Response to AnonReviewer2 [2/2]**
>
>
> ***(R2.Q4) Need of the KL terms in (5). “...What would happen if these KL terms in (5) are dropped and one simply uses SGVB to optimise the resulting loss without the need for discriminators?...”
>
> We agree with the reviewer that modelling density ratios with logistic regression can be problematic and should be avoided whenever possible. However, the use of the discriminators in our method is crucial for a main reason. As stated by the reviewer, using only the reconstruction terms over the latent variables forces the model to encode information into e. However, this information does not need to be related with the target factors (i.e, the ones that are not present in the reference set of images). More concretely, the model can learn to encode most of the information into z and place “reconstructable” but non-relevant information into e. This is clearly avoided when using the discriminator d(x,z) because neutral images generated from p(x|z,e^r)p(z) are forced to be similar to real “reference images”. As a consequence, z can not contain information about target factors, which must be encoded into e.
>
> In order to gain more insights about this issue, we have conducted the suggested ablation study by removing the discriminators of sRB-VAE during training. By following the same experimental setup described in Sec. 5.3, the average performance of the ablated model according to the metrics shown in Table 1 are .371 and .202 for the AffectNet and MNIST respectively. Note that these results are much worse than the ones obtained with our proposed model sRBD-VAE. By visually inspecting the generated samples, we have observed that manipulating the vector e does not significantly modify the images in terms of the target factor. As previously discussed, this shows that the use of the reconstruction losses over the latent space is not enough by itself in order to solve the reference-based disentanglement problem.
>
> To conclude, the reviewer is also referred to our response to Reviewer 1 (see R3.Q1) where we describe another advantage of using the discriminator d(x,z,e). We have added a discussion about these issues in subsection “Optimization via Adversarial Learning” of the revised paper.
>
>
>
> ***(R2.Q4) “why not learn the likelihood variance lambda?“
>
>
> In preliminary experiments we tried to optimize the lambda parameter during training. However, we found that at the early stages of learning, the model tended to assign a very small weight to the reconstruction loss and focus too much on the adversarial component of the loss. We solved this issue by fixing lambda.

---

> ### Author Response · Authors · 2018-11-27
> **Thank you for updating your review**
>
> We are glad to see that the reviewer has considered to update his/her score after reading our rebuttal and paper revision. Please, let us know if there is any remaining clarification that you would like us to provide.

---

### Official Review · AnonReviewer3 · 2018-11-02

**Rating:** 6
**Confidence:** 3

**Review:**

The paper proposes reference based VAEs, which considers learning semantically meaningful feature with weak supervision. The latent variable contains two parts, one related to the reference set and the other irrelevant. To prevent degenerate solutions, the paper proposed to use reverse KL resulting in a ALICE-style objective. The paper demonstrates interesting empirical results on feature prediction, conditional image generation and image synthesis.

I don’t really see how Equation (5) in symmetric KL prevents learning redundant z (i.e. z contains all information of e). It seems one could have both KL terms near zero but also have p(x|z, e) = p(x|z)? One scenario would be the case where z contains all the information about e (which learns the reference latent features), so we have redundant information in z. In this case, the learned features e are informative but the decoder does not use e anyways. To ensure that z does not contain information about e, one could add an adversarial predictor that tries to predict e from z. Note that this cannot be detected by the feature learning metric because it ignores z for RbVAE during training.

The experiments on conditional image generation look interesting, but I wonder if the ground truth transformation for MNIST can be simply described as in some linear transformation on the original image. I wonder if the proposed method works on SVHN, where you can use label information as reference supervision. Moreover, I wonder if it is possible to use multiple types of reference images, but fewer images in each type, to reach comparable or even better performance.

Minor points:
- Why assume that the reference distribution is delta distribution whose support has measure zero, instead of a regular Gaussian?
- (6), (8), (10) seems over complicated due to the semi-supervised nature of the objective. I wonder if having an additional figure would make things clearer.
- Maybe it is helpful to cite the ALICE paper (Li et al) for Equation (10).
- Table 1, maybe add the word “respectively” so it is clearer which metric you use for which dataset.
- I wonder if it is fair enough to compare feature prediction with VAE and other models since they do not use any “weak supervision”; a fairer baseline could consider learning with the weak supervision labels (containing the information that some images have the same label). The improvement on AffectNet compared to regular VAE does not look amazing given the additional weak supervision.

---

> ### Author Response · Authors · 2018-11-12
> **Response to AnonReviewer3 [1/2]**
>
> We thank the reviewer for his detailed feedback. Following, we address his concerns.
>
>
> ***(R3.Q1) “I don’t really see how Equation (5) in symmetric KL prevents learning redundant z (i.e. z contains all information of e)”...”To ensure that z does not contain information about e, one could add an adversarial predictor that tries to predict e from z”
>
> We thank the reviewer for arising this question. We didn’t think about this potential “degenerate” solution before (was not observed in our experiments) and we have concluded that our model naturally avoids the case where redundant information from e is encoded into z. The rationale is as follows. Note that the classifier d(x,z,e) is trained in order to discriminate triplets {x,z,e} obtained from the distributions q(z,e|x)(x) and p(x|z,e)p(z)p(e). On the other hand, the model encoder and generator try to make these distributions as similar as possible. Consider the scenario where a latent z sampled from q(z|x) contains (redundant) information about a sample e generated from q(e|x). In this case, z and e would be conditionally dependent. In contrast, latent variables e and z generated by p(z)p(e) are independent (given that the priors are defined by an isotropic Gaussian distribution). Therefore, our model is penalized in this case since the the discriminator d(x,z,e) would easily differentiate between both distributions  (by exploiting the dependency present in  q(z,e|x) but not in p(z)p(e)). Interestingly, note that the “adversarial predictor” suggested by the reviewer is already implicitly implemented by the discriminator d(x,z,e). We have discussed this issue in the updated version (before last paragraph of Sec 4.3 “Optimization via Adversarial Learning”).
>
>
>
> ***(R3.Q2) “I wonder if the ground truth transformation for MNIST can be simply described as in some linear transformation on the original image...“
>
> The transformations applied to the MNIST datasets are: (i) Colorization, (ii) Modification of the stroke width and (iii) Resizing + zero-padding. Apart from (i), (ii) and (iii) are not linearly dependent on the transformation parameter.
>
>
>
> ***(R3.Q3) “I wonder if the proposed method works on SVHN, where you can use label information as reference supervision...“
>
> Note that our main motivation is to learn a disentangled representation without explicit labelling of the underlying target factors. Using the SVHN as suggested (i.e, considering the digit labels) would imply that the factors of interest are annotated and, therefore, that full or semi-supervision is provided during training. We would like to emphasize that  we are focused in the weakly-supervised setting, where explicit annotations are not needed in order to disentangle the target factors.
>
>
> ***(R3.Q4) “ I wonder if it is possible to use multiple types of reference images ...“
>
> If we understand correctly, in the described setting each reference set would contain images with a specific set of “constant” factors . In this case, we think that our model could easily adress the suggested scenario by splitting the latent variables into more than two subsets and using different discriminators for each reference distribution.
>
> If the reviewer has a concrete idea about a potential scenario where this setting is interesting, we would be grateful to know it. We are currently exploring potential extensions and applications of our proposed model for future work.
>
>
>
> ***(R3.Q5) “ Why assume that the reference distribution is delta distribution whose support has measure zero, instead of a regular Gaussian?“
>
> Using a “delta shaped” prior over latents e allows us to model the assumption that variation factors are constant across reference images. In contrast, note that for unlabelled images the prior p(e) is indeed modelled as a regular Gaussian.
>
>
>
> ***(R3.Q6) “(6), (8), (10) seems over complicated due to the semi-supervised nature of the objective. I wonder if having an additional figure would make things clearer...“
>
> Thank you for the suggestion. We have added Fig. 5 in the Appendix B in order to clarify the formulation and illustrate the training process.
>
>
>
> ***(R3.Q7) “Maybe it is helpful to cite the ALICE paper (Li et al) for Equation (10). Table 1, maybe add the word “respectively” so it is clearer which metric you use for which dataset...“
>
> Suggested changes are added in the updated version of the paper.

---

> ### Author Response · Authors · 2018-11-12
> **Response to AnonReviewer3 [2/2]**
>
>
>
> ***(R3.Q8) “...a fairer baseline could consider learning with the weak supervision labels (containing the information that some images have the same label)...”
>
> We would like to emphasize that the reference-based setting is different from the scenario  where information about samples sharing the same target factors is available. In particular, note that in our case, we only know that reference samples share the same label. In contrast, for the unlabelled distribution we do not have access to this information (e.g what faces share the same expression). This is because our main goal is to avoid the explicit annotation of the factors of interest. As discussed in the related work, assuming supervision in terms of images sharing the same label have been previously considered in previous approaches [Mathieu et al., 2016; Donahue et al., 2018]. However, in “reference-based disentangling” this type of supervision is not available during training and, to the best of our knowledge, no previous methods are able to naturally address this problem. Our comparison with unsupervised models is intended to show that our model is able to exploit the weak-supervision provided by the reference set and its advantages. However, as suggested by Reviewer 3 , we have evaluated the method presented  in [Mathieu et. al, 2016] by adapting it to our “reference-based” setting.  Note that this method also can exploit the weak-labels provided in the reference-set. Please, see R2.Q1 for a detailed discussion.

---

### Official Review · AnonReviewer1 · 2018-11-07
**Interesting approach, somewhat artificial setup, limited interpretation of "disentangling representation learning"**

**Rating:** 7
**Confidence:** 4

**Review:**

The authors address the problem of representation learning in which data-generative factors of variation are separated, or disentangled, from each other. Pointing out that unsupervised disentangling is hard despite recent breakthroughs, and that supervised disentangling needs a large number of carefully labeled data, they propose a “weakly supervised” approach that does not require explicit factor labels, but instead divides the training data in to two subsets. One set, the “reference set” is known to the learning algorithm to leave a set of generative “target factors” fixed at one specific value per factor, while the other set is known to the learning algorithm to vary across all generative factors. The problem setup posed by the authors is to separate the corresponding two sets of factors into two non-overlapping sets of latents.

Pros:

To address this problem, the authors propose an architecture that includes a reverse KL-term in the loss, and they show convincingly that this approach is indeed successful in separating the two sets of generative factors from each other. This is demonstrated in two different ways. First, quantitatively on an a modified MNIST dataset, showing that the information about the target factors is indeed (mostly) in the set of latents that are meant to capture them. Second, qualitatively on the modified MNIST and on a further dataset, AffectNet, which has been carefully curated by the authors to improve the quality of the reference set. The qualitative results are impressive and show that this approach can be used to transfer the target factors from one image, onto another image.

Technically, this work combines and extends a set of interesting techniques into a novel framework, applied to a new way of disentangling two sets of factors of variation with a VAE approach.

Cons:

The problem that this work solves seems somewhat artificial, and the training data, while less burdensome than having explicit labels, is still difficult to obtain in practice. More importantly, though, both the title and the start of the both the abstract and the introduction are somewhat misleading. That’s because this work does not actually address disentangling in the sense of “Learning disentangled representations from visual data, where high-level generative factors correspond to independent dimensions of feature vectors…” What it really addresses is separating two sets of factors into different parts of the representation, within each of which the factors can be, are very likely are, entangled with each other.

Related to the point that this work is not really about disentangling, the quantitative comparisons with completely unsupervised baselines are not really that meaningful, at least not in terms of what this work sets out to do. All it shows is whether information about the target factors is easily (linearly) decodable from the latents, which, while related to disentangling, says little about the quality of it. On the positive side, this kind of quantitative comparison (where the authors approach has to show that the information exists in the correct part of the space) is not pitted unfairly against the unsupervised baselines.

===
Update:
The authors have made a good effort to address the concerns raised, and I believe the paper should be accepted in its current form. I have increased my rating from 6 to 7, accordingly.

---

> ### Author Response · Authors · 2018-11-12
> **Response to AnonReviewer1 [1/2]**
>
> We would like to thank the reviewer for his useful comments and remarks. Detailed discussion about his specific concerns are addressed below.
>
>
> ***(R1.Q0) Clarification about quantitative results
>
> First of all, we would like to clarify that our experiments also include quantitative evaluation over the AffectNet. From the reviewer's description in the “Pros” section, it could be interpreted that we only provide quantitative results over this dataset.
>
>
>
> ***(R1.Q1) - Practical applications of the learning setting : “..The problem that this work solves seems somewhat artificial...”
>
> We strongly believe that addressing the introduced problem can be useful in different scenarios. One of the motivation of our experiments on the AffectNet was to show a concrete advantage of this type of supervision in a practical case. Note that in facial behavior analysis/synthesis large-scale datasets are typically very hard to annotate. The reason is that facial gestures depend on a combination of a large number of facial muscle activations and their corresponding intensities (i.e. Action Units) [Ekman, 1997]. Therefore, fine-grained annotation of facial gestures is very tedious and require expert coders. By contrast, collecting a large data set of neutral faces is much easier and can be carried out by non-expert annotators.  Another interesting application that we plan to explore in future work is “weakly-supervised” artistic-style disentangling. In this case, we will consider the unlabelled dataset to be a collection of paintings (containing a large-number of styles that do not need to be labelled). On the other hand, we will consider the reference samples as images with a “constant” style (real photographs). Note that in this case, the reference dataset would be almost free to collect. By training our model on this data, we would be able to learn a latent representation of the painting styles with no supervision and manipulate it in order to transfer styles, interpolate them or synthetically generate new ones.  Following the same idea, another potential application where the reference-based supervision could be useful is automatic colorization of grayscale photographs. In this case, multiple colorizations for the same picture could be synthetized by injecting random noise into the latent variable e. Note that in this scenario, the reference images would be obtained by removing the color of natural images (forming the unlabelled set) . Again, in this application the reference-set would be very easy to collect.

---

> > ### Author Response · Authors · 2018-11-24
> > **Response to AnonReviewer1 [2/2]**
> >
> >
> > ***(R1.Q2) Definitions of disentangled representations.
> >
> > We believe that reviewer’s concern is caused by a different interpretation of “disentanglement” compared to ours. If we have correctly understood, the reviewer refers to a specific definition of disentangled representation implying a bijective mapping between one generative factor and a single dimension of a the latent representation (i.e, feature vector). Despite the fact that this definition has been adopted by recent unsupervised approaches [Higgins et. al, 2017, Kumar et. al, 2018] focusing on the disentanglement of simple generative factors, we think that this view is not appropriate for more challenging problems. For example, it is unrealistic to expect that a high-level factor such as the facial expression can be modelled by a unique continuous value.
> >
> > In our work, we adopt a more flexible interpretation of “disentanglement”, where the information of a complex high-level factor of variation (e.g the digit style) can be encoded into a subset of dimensions of the latent representation (i.e, the vector e in our model). Note that we can consider this complex generative factor to be a composition of simpler transformations (e.g, color, size, width) which, indeed, can be entangled in the vector e.
> >
> > In this scenario, the disentanglement arise from the fact that the rest of factors non-related with the style are encoded into a separate set of dimensions of the latent representation  (i.e, the vector z). Under this assumptions, our definition of disentangling is coherent with reviewer’s statement: “what it really addresses is separating two sets of factors into different parts of the representation”. In fact, we think that the word “separating” could be replaced by “disentangling” without modifying the implications. Note that our notion of disentangled representation has been previously employed in other works where a complex high-level generative factor  is disentangled from the rest (e.g. face identity in [Donahue et al., 2018]). Being said this, we agree that the sentence : “Learning disentangled representations from visual data, where high-level generative factors correspond to independent dimensions of feature vectors…” can be misleading. For this reason, we have rephrased it in the updated version of the paper.
> >
> >
> >
> > ***(R1.Q3) Evaluation procedure actually measuring disentanglement.
> >
> > Following the discussed interpretation of disentangled representation, we think that the followed evaluation procedure is appropriate in order to effectively measure the level of “disentanglement”. This is because, as stated by the reviewer, our model “has to show that the information exists in the correct part of the space” (i.e, in the latent variable e and not in z) and, therefore, that the target factors are disentangled from the rest.

---

### Author Response · Authors · 2018-11-12
**General response to the reviewers**

We thank all the reviewers for their constructive feedback which, honestly, is being very useful in order to improve the quality of our work. We have uploaded a revised version of our paper where we have incorporated additional material and suggestions. The main changes are summarised as follows:

***Sec. 1:
-Added discussion about the advantages of reference-based supervision in facial expression analysis/synthesis (AnonReviewer1)
- Rephrased definition of disentangled representations. (AnonReviewer1)
***Sec. 4
- Added discussion about the need of using discriminators in our model (AnonReviewer2 & AnonReviewer3)
***Experiments
- Added comparison and discussion with [Mathieu, et al,2016] and [Chen,et al 2018] (AnonReviewer2)
***Appendix
- Figure added to clarify the model and training procedure (AnonReviewer3)
***Added suggested references by the reviewers and other minor comments


More detailed discussion about these and other issues is provided to each reviewer independently. We hope that this helps to address the reviewer’s concerns and, if considered, raise their final scores. Please, do not hesitate to ask for more clarifications if needed.

---

### Meta-Review · Area_Chair1 · 2018-12-14

**Confidence:** 4
**Recommendation:** Reject

**Metareview:**

This is a proposed method that studies learning of disentangled representations in a relatively specific setting, defined as follows: given two datasets, one unlabeled and another that has a particular factor of variation fixed, the method will disentangle the factor of variation from the others. The reviewers found the method promising, with interesting results (qual & quant).

The weaknesses of the method as discussed in the reviews and after:

- the quantitative results with weak supervision are not a big improvement over beta-vae-like methods or mathieu et al.
- a red flag of sorts to me is that it is not very clear where the gains are coming from: the authors claim to have done a fair comparison with the various baselines, but they introduce an entirely new encoder/decoder architecture that was likely (involuntarily, but still) tuned more to their method than others.
- the setup as presented is somewhat artificial and less general than it could be (however, this was not a major factor in my decision). It is easy to get confused by the kind of disentagled representations that this work is aiming to get.

I think this has the potential to be a solid paper, but at this stage it's missing a number of ablation studies to truly understand what sets it apart from the previous work. At the very least, there is a number of architectural and training choices in Appendix D -- like the 0.25 dropout -- that require more explanation / empirical understanding and how they generalize to other datasets.

Given all of this, at this point it is hard for me to recommend acceptance of this work. I encourage the authors to take all this feedback into account, extend their work to more domains (the artistic-style disentangling that they mention seems like a good idea) and provide more empirical evidence about their architectural choices and their effect on the results.

---

> ### Author Response · Authors · 2019-01-23
> **Response to the AC metareview overriding reviewer consensus**
>
> We would like to thank again all the reviewers and the area chair for their participation during the reviewing period. We honestly believe that it significantly contributed to improve our work. Being said that and given the nature of OpenReview, we would like to provide a response to the AC metareview which resulted in a reject decision. We think that his/her arguments were too limited to warrant over-riding the reviewer consensus for acceptance. In summary, from our point of view, the reasons appear vague, arbitrary in some aspects and/or were never raised during the reviewing period:
>
>
>
> 1 (AC) “the quantitative results with weak supervision are not a big improvement over beta-vae-like methods or Mathieu et al.”
>
> → It is not clear how “big improvements” should be for the AC not to be used as a reason to reject a paper. Experimentally, we find that our method improves over all the compared state-of-the-art methods on the evaluated datasets. The comparison with Mathieu et. al was added according to a suggestion by AnonReviewer2, who raise his/her score based on the provided results. Additionally, we show how our model can be applied to different problems (conditional image generation and attribute transfer) which can not be addressed using the compared methods.
>
>
>
> 2 (AC) “a red flag of sorts to me is that it is not very clear where the gains are coming from: the authors claim to have done a fair comparison with the various baselines, but they introduce an entirely new encoder/decoder architecture that was likely (involuntarily, but still) tuned more to their method than others.”
>
> → This point was not raised in any of the reviews and, therefore, we had not the opportunity to clarify it. As described in the paper, our networks are based on a standard conv/deconv networks with upsampling and downsampling operations. This is the most standard architecture used in VAE-based models and GAN literature. Moreover, our architecture uses exactly the same building blocks as [Karras et. al, 2018]. We think that characterising this as an “entirely new encoder/decoder architecture” is unreasonable. We did not explore other types of architecture designs and all the evaluated methods used the same hyper-parameters without any tuning.
>
>
>
> 3 (AC) "the setup as presented is somewhat artificial and less general than it could be (however, this was not a major factor in my decision). It is easy to get confused by the kind of disentangled representations that this work is aiming to get."
>
> → This point was raised by AnonReviewer1 during the revision. We clarified our concept of disentanglement and discussed different potential applications of our setup. We updated our paper accordingly. The reviewer was convinced by our response and raised his score to (Good paper, accept). No additional comments on this issue are provided by the AC.
>
>
>
> 4 (AC) "I think this has the potential to be a solid paper, but at this stage it's missing a number of ablation studies to truly understand what sets it apart from the previous work. At the very least, there is a number of architectural and training choices in Appendix D -- like the 0.25 dropout -- that require more explanation / empirical understanding and how they generalize to other datasets."
>
> --> No description about the “number of ablation studies” is provided. A single concrete point is raised here: that we do not provide data on how the used dropout-rate of 0.25 generalizes to other datasets. First of all, we did not fine-tune this hyper-parameter given that we found that the default value worked well in all our experiments. Moreover, the lack of in-depth evaluation of how the used dropout rates generalize to other datasets seems plentiful across the papers in ICLR/ICML/NIPS/CVPR/ICCV/ECCV that use dropout strategies, and does not appear to be a commonly accepted reason to reject papers from publication. Finally, it is worth to clarify that the “dropout strategy” is only used in our model (because it is only applied when the discriminator uses features as input) and therefore, the results of the other baselines are not affected by this hyper-parameter.